

# The early Miocene balaenid *Morenocetus parvus* from Patagonia (Argentina) and the evolution of right whales

Mónica R. Buono[1], Marta S. Fernández[2], Mario A. Cozzuol[3],
José I. Cuitiño[1] and Erich M.G. Fitzgerald[4,5,6]

[1] Instituto Patagónico de Geología y Paleontología, CCT CONICET-CENPAT, Puerto Madryn, Chubut, Argentina
[2] División Paleontología Vertebrados, Unidades de Investigación Anexo Museo, Facultad de Ciencias Naturales y Museo, UNLP, La Plata, Buenos Aires, Argentina
[3] Departamento de Zoologia, Instituto de Ciências Biológicas, Universidade Federal de Minas Gerais, Belo Horizonte, Minas Gerais, Brazil
[4] Geosciences, Museums Victoria, Melbourne, Victoria, Australia
[5] Department of Vertebrate Zoology, National Museum of Natural History, Smithsonian Institution, Washington, DC, USA
[6] Department of Life Sciences, Natural History Museum, London, UK

Corresponding author
Mónica R. Buono,
buono@cenpat-conicet.gob.ar

## ABSTRACT

Balaenidae (right and bowhead whales) are a key group in understanding baleen whale evolution, because they are the oldest surviving lineage of crown Mysticeti, with a fossil record that dates back ~20 million years. However, this record is mostly Pliocene and younger, with most of the Miocene history of the clade remaining practically unknown. The earliest recognized balaenid is the early Miocene *Morenocetus parvus Cabrera, 1926* from Argentina. *M. parvus* was originally briefly described from two incomplete crania, a mandible and some cervical vertebrae collected from the lower Miocene Gaiman Formation of Patagonia. Since then it has not been revised, thus remaining a frequently cited yet enigmatic fossil cetacean with great potential for shedding light on the early history of crown Mysticeti. Here we provide a detailed morphological description of this taxon and revisit its phylogenetic position. The phylogenetic analysis recovered the middle Miocene *Peripolocetus* as the earliest diverging balaenid, and *Morenocetus* as the sister taxon of all other balaenids. The analysis of cranial and periotic morphology of *Morenocetus* suggest that some of the specialized morphological traits of modern balaenids were acquired by the early Miocene and have remained essentially unchanged up to the present. Throughout balaenid evolution, morphological changes in skull arching and ventral displacement of the orbits appear to be coupled and functionally linked to mitigating a reduction of the field of vision. The body length of *Morenocetus* and other extinct balaenids was estimated and the evolution of body size in Balaenidae was reconstructed. Optimization of body length on our phylogeny of Balaenidae suggests that the primitive condition was a relatively small body length represented by *Morenocetus*, and that gigantism has been acquired independently at least twice (in *Balaena mysticetus* and *Eubalaena* spp.), with the earliest occurrence of this trait in the late Miocene–early Pliocene as represented by *Eubalaena shinshuensis*.

## INTRODUCTION

Balaenidae have been considered a key group in understanding baleen whale (Mysticeti) evolution, because they are the oldest surviving lineage of crown Mysticeti, as suggested by their ancient stratigraphic occurrence (*Cabrera, 1926*) coupled with an early divergence date estimated from mtDNA data (*McGowen, Spaulding & Gatesy, 2009*; *Marx & Fordyce, 2015*). Despite their long evolutionary history, living balaenids exhibit relatively low taxonomic richness of four species in two genera (the bowhead whale *Balaena* and three right whales *Eubalaena* spp.)—in comparison to at least eight extant balaenopterid species—and share conservative morphology distinguishing them from other mysticetes, including an extremely large head (about one-third the body length), highly arched rostrum with long baleen plates (in right whales present about 2 m–3 m while in bowhead whales are longer with a range between 3 and 4 m), huge dorsally bowed lower lips and lack of ventral throat grooves (*Cummings, 1985*; *Reeves & Leatherwood, 1985*). Many of these morphological features have been functionally linked to the continuous ram filter (or skim) feeding employed by all balaenids (*Bouetel, 2005*; *Lambertsen et al., 2005*). Recent phylogenetic analyses of morphological and molecular data suggest that the origins of extant mysticete families lie between the late Oligocene and early Miocene (~28–16 Ma) (*Steeman et al., 2009*; *Marx & Fordyce, 2015*). Yet fossil representatives of the living families Balaenopteridae (rorquals), Eschrichtiidae (grey whale), Neobalaenidae (pygmy right whale) and Balaenidae (right whales) are virtually unknown from this interval. The singular exception to this pattern is the early Miocene balaenid *Morenocetus parvus Cabrera, 1926* from Patagonia, Argentina. Almost all known fossil balaenids are late Miocene or younger in age (e.g. *Bisconti, 2003*; *2005*; *Fitzgerald, 2004*), hence *Morenocetus* establishes a significant earlier history for Balaenidae dating back ~20 Ma. Although a late Oligocene (~28 Ma) stem balaenid was reported from New Zealand (*Fordyce, 2002*), it has not been formally described yet, and recent analyses recovered it outside of Balaenidae (*Marx & Fordyce, 2015*; *Marx, Bosselaers & Louwye, 2016*; *Tsai & Fordyce, 2016*; *Marx & Kohno, 2016*; *Marx, Bosselaers & Louwye, 2016*). This makes *Morenocetus* the geologically earliest confirmed member of an extant mysticete family (*Bisconti, 2005*; *Marx & Fordyce, 2015*; *Gol'din & Steeman, 2015*), and therefore a pivotal fossil calibration for mysticete molecular clock divergence estimates. Furthermore, being the earliest balaenid, *Morenocetus* may provide insights into the primitive morphology of Balaenidae and thus shed light on the contentious higher-level relationships within crown Mysticeti; namely, whether the highly specialized Neobalaenidae are more closely related to Balaenidae or Balaenopteroidea (Balaenopteridae + Eschrichtiidae).

*Cabrera's (1926)* original description of *M. parvus* was based on two incomplete crania, a mandible and some cervical vertebrae collected from the lower Miocene Gaiman Formation exposed at the Cerro Castillo locality (Chubut province, Patagonia, Argentina). The original descriptions of both the holotype and referred specimen are brief and do not document phylogenetically significant regions of the cranium such as the

basicranium and periotic. This gap in anatomical data on *M. parvus* is reflected in disparate estimates of this key taxon's phylogenetic position within Balaenidae (*Bisconti, 2005*; *Churchill, Berta & Deméré, 2012*; *Bisconti, Lambert & Bosselaers, 2017*). Although *M. parvus* has been included in comprehensive analyses of mysticete phylogeny (*Bisconti, 2000*; *Churchill, Berta & Deméré, 2012*; *Fordyce & Marx, 2013*; *Tsai & Fordyce, 2015*; *Marx & Fordyce, 2015*; *Gol'din & Steeman, 2015*; *Bisconti, Lambert & Bosselaers, 2017*), their character coding was based solely on *Cabrera's (1926)* illustrations and brief description. In the latter studies, there was limited consideration of periotic morphology, and character data from the informative referred specimen have never been incorporated in analyses. In the latest analysis *Morenocetus* was recovered in a clade with *Balaenella* + *Balaena* and sister to *Balaenula* + *Eubalaena*, respectively (*Bisconti, 2005*), outside a Balaenidae + Neobalaenidae clade (*Bisconti, Lambert & Bosselaers, 2017*) or in an unresolved position recovered by *Churchill, Berta & Deméré (2012)*. The early stratigraphic occurrence of *Morenocetus* hints at a basal position in balaenid phylogeny, which has been corroborated by some recent analyses (*Fordyce & Marx, 2013*; *Tsai & Fordyce, 2015*; *Marx & Fordyce, 2015*). Nonetheless, *Gol'din & Steeman (2015)* instead recovered the middle Miocene *Peripolocetus* as the earliest diverging balaenid.

Here we describe the anatomy of *M. parvus* in depth, based on the type and referred specimens, and revise the geologic setting and age of this important fossil mysticete. Our thorough anatomical description forms the basis for the most complete character analysis of *Morenocetus* to-date and a new phylogenetic analysis of Balaenidae. The results of our analysis support the basal position of *Morenocetus* within Balaenidae, which sheds light on the evolutionary history of right whales.

## MATERIALS AND METHODS

### Materials

The re-description of *M. parvus* is based on the holotype (MLP 5–11) and referred specimens (MLP 5–15) described originally by *Cabrera (1926)* and deposited in the Vertebrate Palaeontology collection of the Museo de La Plata. In addition, *Cabrera (1926)* referred other material to *M. parvus*: four cervical vertebrae (MLP 5–30) and a mandible (MLP 5–21). The cervical vertebrae (MLP 5–30) are not included herein as they could not be located in the La Plata Museum collection. Bone characteristics of the mandible (MLP 5–21) (i.e., lower weight and signals of little mineral replacement) clearly differs from MLP 5–11 and 5–15, indicating that it could correspond to a much more recent fossil or even extant balaenid. In order to test if the mandibles could correspond to *M. parvus*, a collagen and nitrogen analysis were carried out. High concentrations of collagen and nitrogen of the MLP 5–21 (0.075% $N$ 0.40% collagen in the holotype; 1.79% $N$ 9.59% collagen in MLP 5–21) strongly suggest that MLP 5–21 is not Miocene in age. In consequence, MLP 5–21 is not referable to *M. parvus* and we do not consider it further here. Although the exact age of the specimen cannot be determined at this stage, its collagen concentration and minimal mineral replacement strongly suggest that it dates from the Pleistocene–Holocene.

The balaenid specimens used in the comparative and phylogenetic analyses are listed in the Supplemental Information.

## Methods

### Collagen and nitrogen analysis

To determinate nitrogen content of the bone samples of MLP 5–21 and MLP 5–11 we used the Kjeldahl method (*Kjeldahl, 1883*). In this method, the nitrogen of the sample is converted to ammonium by reaction with sulfuric acid. The content of ammonium is determined from the amount of ammonia liberated by distillation with a strongly alkalized solution. The collagen concentration is established using the following equation: Protein (%) = % $N \times 5.36$.

### Preparation methods and photographs

The specimens MLP 5–11 and MLP 5–15 were prepared in the Museo de La Plata using pneumatic chisels and hand tools. The left ear bone of MLP 5–11 was prepared under magnifying glass but it was not disarticulated from the cranium to avoid its destruction. Photographs were taken with a Nikon D3000 camera and a 55 mm lens.

### Anatomical description

The skull anatomical terminology follows that proposed by *Mead & Fordyce (2009)* with additions from *Fordyce & Marx (2013)*, *Marx, Bosselaers & Louwye (2016)*, for those skull anatomical terms more specific for mysticetes (squamosal cleft, squamosal prominence) and *Boessenecker & Fordyce (2015)* for bulla anatomical terms (lateral and medial lobe). Most of the measurements were taken following the standard protocols of *Perrin (1975)*. The description is mainly based on the left or right side of the cranium, whichever is more completely preserved, on the holotype. The features not observable on the holotype and/or variable in the two known specimens are indicated. The periotic was described in situ and description focused on the left side. Computed tomography techniques was used to generate a 3D reconstruction and coronal sections of the skull of the holotype of *M. parvus* to study internal sutures and endocranial features.

Nine calf specimens of southern right whales (*Eubalaena australis*) that stranded near the nursery grounds at Península Valdes, Chubut province, Argentina, were dissected to analyze soft tissue structures (Table S1). Disecctions were approved by the Dirección de Fauna y Flora Silvestre del Chubut (approval number 30/2010).

### Physical maturity

To estimate the ontogenetic stage of the specimens we analyzed the degree of cranial suture closure, according to the rating system proposed by *Walsh & Berta (2011)*; bony texture (e.g., punctuate texture of occipital condyles); and development of the skull crests. The cranial suture closure system originally proposed by *Walsh & Berta (2011)* was not tested for balaenids. Nevertheless, we had access to *Eubalaena australis* skulls at different ontogenetic stages (neonates, calves, juveniles and adults, Table 1) that allowed us to corroborate a similar suture closure pattern in at least this species of Balaenidae.

**Table 1 List of specimens of *Eubalaena australis* analyzed for cranial suture closure.**

| Collection number | Age | BZ (cm) | TL (m) | Cranial suture | State of close suture according to *Walsh & Berta (2011)* |
|---|---|---|---|---|---|
| CNPMAMM 748 | Neonate | 54 | 3.64 | BO/BE | SR1 |
| | | | | BO/EX | SR1 (SR2?) |
| | | | | SO/EX | SR2[a] |
| CNPMAMM 746 | Calf | ~70 | – | BO/BE | SR1 |
| | | | | BO/EX | SR1 |
| | | | | SO/EX | SR1 |
| CNPMAMM 742 | Calf | – | – | BO/BE | SR3 |
| | | | | BO/EX | SR4 |
| | | | | SO/EX | SR4 |
| Without number[b] | Juvenile | – | 11.40 | BO/EX | SR4 |
| USNM 267612 | Adult | 216 | 13.7 | BO/EX | SR4 |
| | | | | SO/EX | SR4 |
| CNPMAMM 774 | Adult | 256 | 14.45 | BO/EX | SR3 (right side)-SR4 (left side) |
| | | | | SO/EX | SR4 |

Notes:
  BE, basisphenoid; BO, basioccipital; BZ, bizygomatic width, TL, total length; EX, exoccipital; SO, supraoccipital.
  [a] The closure of the suture is in a medial direction from the lateral edges.
  [b] Specimen in exhibition in "Centro de Interpretación Istmo Ameghino," Península Valdés, Chubut Province, Argentina.

## Orientation of the skull

The correct anatomical orientation of the skull in mysticetes is an important issue, especially with respect to comparisons between taxa and also for the interpretation of morphological phylogenetic characters. *Yamada et al. (2006)* proposed a standardized protocol to orientate the skull of mysticetes (with focus on balaenopterids) in dorsal view. However, accurate anatomical orientation of the skull in extinct balaenids is challenging because most specimens lack the rostrum. To this it must be added that, due to the arched rostrum and a vertically oriented occipital shield, the condylobasal length of the skull of balaenids is not homologous to other cetaceans. Resolving this problem is essential to determining the configuration of important characters. In order to resolve this topic we propose three anatomical landmarks to orientate the crania of fossil balaenids based on observations and dissections of specimens of *Eubalaena australis* (Fig. 1). We did not perform dissections of specimens of *Eubalaena glacialis*, *Eubalaena japonica* or *Balaena mysticetus*. However, we analyzed osteological specimens of these species (Supplemental file) to test the first and second anatomical landmarks proposed, showing a similar pattern to *Eubalaena australis*. Future dissections of these species could further corroborate the position of the landmarks we have used in this present study and their application.

1. The *position of the nasal fossa*: the nasal fossa, defined as the bony fossa on the dorsal surface of the skull holding the soft structures of the upper nasal passage, epithelium, blowhole and vestibule, is located in the posterodorsal region of the cranium (*Buono et al., 2015*). Detailed anatomical analysis of the nasal complex in *Eubalaena australis* shows that, in dorsal or anterior view, the posteriormost region of the nasal fossa floor

(where the blowhole is located) is parallel to the horizontal plane of the skull (Fig. 1A). This character is also evident in *Balaena mysticetus*, where the skull is strongly arched in a dorsoventral direction, with the posteriormost region of the nasal floor oriented parallel to the horizontal plane of the skull. This character could not be applied to *Morenocetus* because the rostral bones are not preserved.

2. The *orientation of the postglenoid process of the squamosal*: The anatomical exploration of the skull and soft structures associated in *Eubalaena australis*, demonstrated that orientation of the postglenoid process is also an accurate landmark for skull orientation. The postglenoid process is posteroventrally directed, defining an acute angle of approximately 50°–55° (pointing back) with the horizontal plane of the skull (Fig. 1B).

3. The *position of the foramen magnum*: this anatomical point is a significant landmark as it allows orientation of the skull in relation to the vertebral column. Dissections of extant balaenids permit a precise determination of the location and course of the spinal cord within the foramen magnum and its path to the braincase. In Eubalaena australis, the spinal cord leaves the skull through the foramen magnum in a dorsal position and the foramen magnum is deflected ventrally forming an acute angle of approximately 30° with the horizontal plane of the skull (Figs. 1C and 1D).

### Phylogenetic analysis

Phylogenetic analyses were carried out on a combined data matrix based on a recent mysticete dataset (*Marx & Fordyce, 2015*). As our contribution is focused on balaenids, this dataset was modified as follows: unpublished specimens were excluded; the taxonomic sample of balaenopteroids and cetotheriids was reduced; additional balaenid taxa were sampled with the inclusion of the three species of *Eubalaena* (*Eubalaena australis*, *Eubalaena glacialis* and *Eubalaena japonica*); and the list of characters was modified (see the Supplemental file 1 for the complete list of taxa and characters). *Balaenula balaenopsis* was not included in our analysis because the taxonomic status of this taxon needs to be carefully revised. There are serious doubts that all of the skeletal elements referred by *Van Beneden (1872)* to *Balaenula balaenopsis* actually represent one individual, let alone one taxon.

Our thorough anatomical description of *Morenocetus* was used for completing and/or correcting previous character scoring of this taxon. Modifications to the character coding of *Morenocetus* as well as in other balaenids are detailed in Supplemental file 1.

The total-evidence dating analysis was conducted with the addition of the molecular dataset from *McGowen, Spaulding & Gatesy (2009)*, which was pruned to match our taxon sample.

Heuristic parsimony analysis of the dataset was performed in TNT version 1.5 (*Goloboff & Catalano, 2016*) using the traditional search under equal and implied weights ($K = 3$; $K = 6$; $K = 10$). Some characters were treated as ordered (see Supplemental file). The analysis was performed using 1,000 replicates of Wagner trees (using random addition sequences), tree bisection reconnection (TBR) branch swapping holding 10 trees per

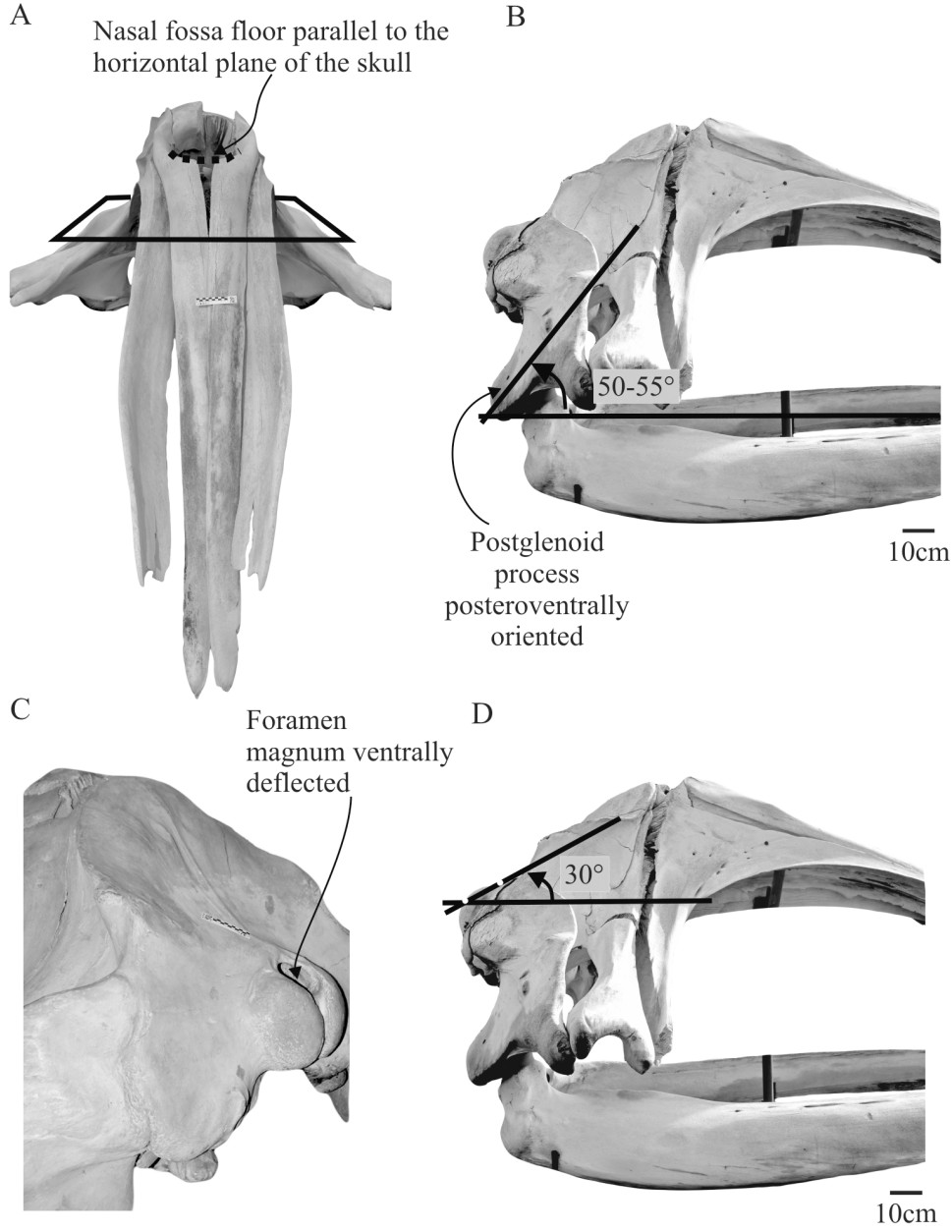

**Figure 1** **Anatomical landmarks proposed to orient the crania of fossil balaenids.** Skull of *Eubalaena australis* in (A) anterior (CNPMAMM 774) (B) lateral (CNPMAMM 774) and (C) posterolateral and (D) lateral view (MACN 54.119, CNPMAMM 774 respectively) showing the three proposed anatomical landmarks to orientate the crania in fossil balaenids.

replicate. The best trees obtained at the end of the replicates were subjected to a final round of TBR branch swapping. The resulting most parsimonious trees (MPTs) were summarized using strict consensus trees with zero-length branches collapsed (i.e., "rule 1" of *Coddington & Scharff, 1994*). Branch support was calculated using the decay index. To identify unstable taxa we used the IterPCR procedure (*Pol & Escapa, 2009*) over the entire set of MPTs. This procedure allows the identification of the set of characters that positively

support alternative positions of the unstable taxon from the set of characters scored with missing entries that may diminish the instability of each taxon (*Pol & Escapa, 2009*; *Escapa & Pol, 2011*).

### Estimation of the body size

In order to estimate the total length (TL) of *M. parvus* and other extinct balaenids, we used two alternative methods of estimating body length: 1) the regression equation proposed by *Pyenson & Sponberg (2011)* for stem mysticetes: [log (TL) = 0.92 * (log (BZW) − 1.72) + 2.68] based on the bizygomatic width (BZW). We decided to use this equation, despite not being specific for balaenids, because *Pyenson & Sponberg (2011)* developed an alternative approach using phylogenetic relationships and multiple cranial metrics to address this issue. Application of this approach significantly increased the accuracy of reconstructed body length in Neoceti. However, because *Pyenson & Sponberg (2011)* analysis did not include balaenids in their dataset, we applied the regression equation of *Lambert et al. (2010)*, which includes extant balaenids in their dataset: [*y* (TL) = 8.209 * *x* (BZW) + 66.69]. In *Morenocetus* BZW is not completely preserved in any of the referred specimens; thus the TL could be underestimated. For comparative purpose we estimated the TL in the balaenids, of which the BZW was available from literature: *Balaenula astensis* (BZW = 700 mm; *Bisconti, 2000*), *Balaena montalionis* (estimated BZW = 820 mm from *Bisconti, 2000*, Table 2 p. 41), *Balaenella brachyrhynus* (estimated BZW = 900 mm, the skull is broken at this level so this measure is an underestimate), *Eubalaena shinshuensis* (estimated BZW = 1,490 mm, *Kimura, 2009*) and *Eubalaena ianitrix* (BZW = 1,660 mm; *Bisconti, Lambert & Bosselaers, 2017*).

The TLs were optimized onto the cladogram obtained here using a modified version of the body size classes proposed by *Fitzgerald (2010)*: small body size (<6 m long), large body size (6–12 m long), and very large body size (>12 m long).

## GEOLOGICAL SETTING

The holotype of *M. parvus* was collected from the "...El Castillo in front of Trelew locality..." (sic) "...in the marine Patagonian Formation..." (*Cabrera, 1926*: 364). The El Castillo locality is located about 10 km south of Trelew city, in the southern margin of the Lower Valley of the Chubut River, Chubut province (Fig. 2). Along the southern margin of this valley a 200 m thick, subhorizontal sedimentary succession is exposed, comprising Paleogene–Neogene continental and marine units (*Scasso & Castro, 1999*; *Scasso & Bellosi, 2004*), including the "marine Patagonian Formation" mentioned by *Cabrera (1926*: 364). The latter corresponds to an informal unit also known as Patagoniense, rooted in the literature since the nineteenth century (*Ameghino, 1906*; *Frenguelli, 1935*; *Simpson, 1935*; *Feruglio, 1949*). Currently it comprises several lithostratigraphic units in Patagonia, ranging in age from the late Oligocene to the early Miocene (*Cuitiño et al., 2017*). These units were all deposited in shallow marine to estuarine environments, during a major marine transgression that flooded a large part of Patagonia (*Scasso & Castro, 1999*; *Malumián & Náñez, 2011*; *Cuitiño et al., 2017*). The Patagoniense beds that crop out in the Lower Valley of the Chubut River and along the

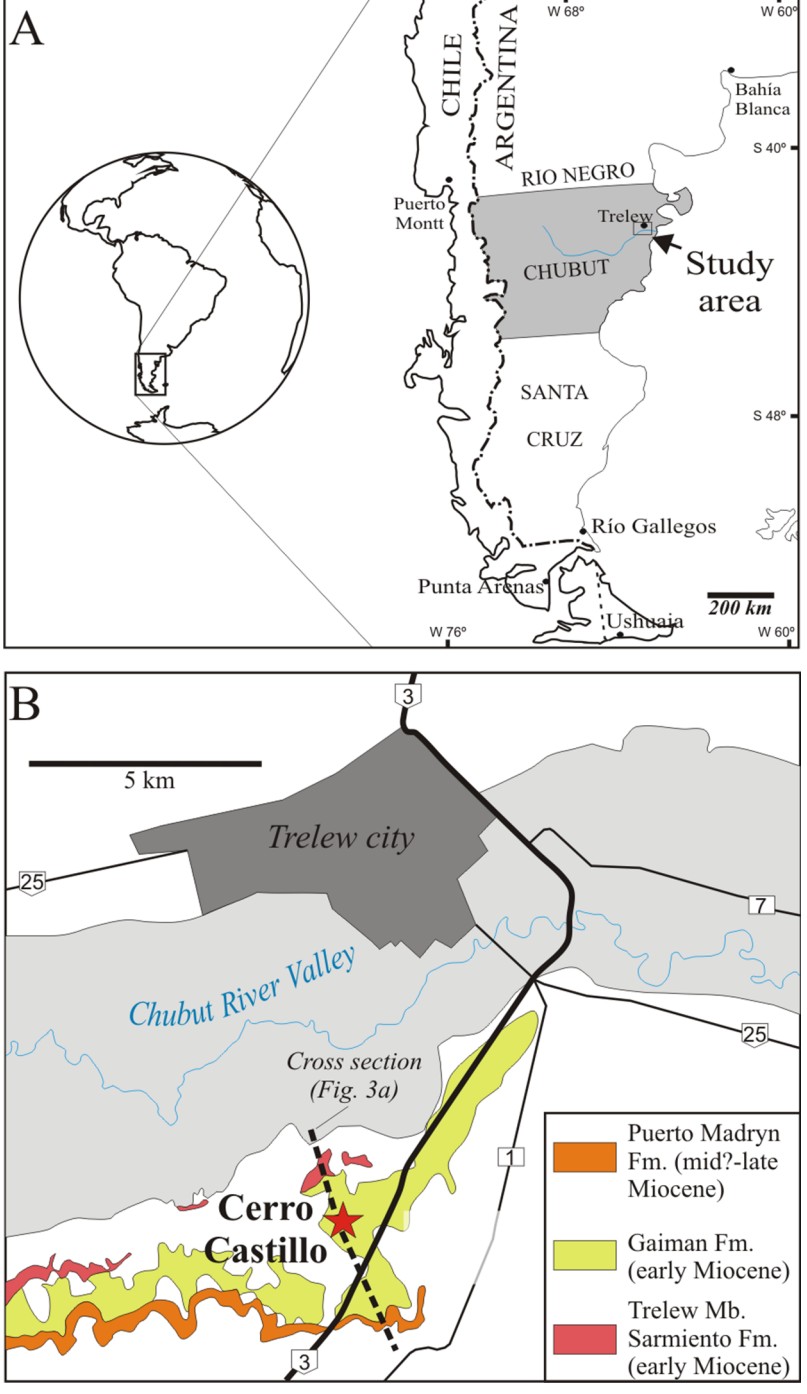

**Figure 2 Locality map for *Morenocetus parvus.*** (A) Regional map showing the location of the study area. (B) Type locality (red star) for the holotype and referred specimens of *Morenocetus parvus*. Only Neogene sedimentary units are mapped.

coastal cliffs of eastern Chubut Province were assigned to the Gaiman Formation (*Mendía & Bayarsky, 1981*). In Cerro Castillo this unit is nearly 100 m thick (Fig. 3) and unconformably overlies the Trelew Member of the Sarmiento Formation, composed of continental yellowish to whitish-grey, sandy tuffs and tuffs with terrestrial mammals

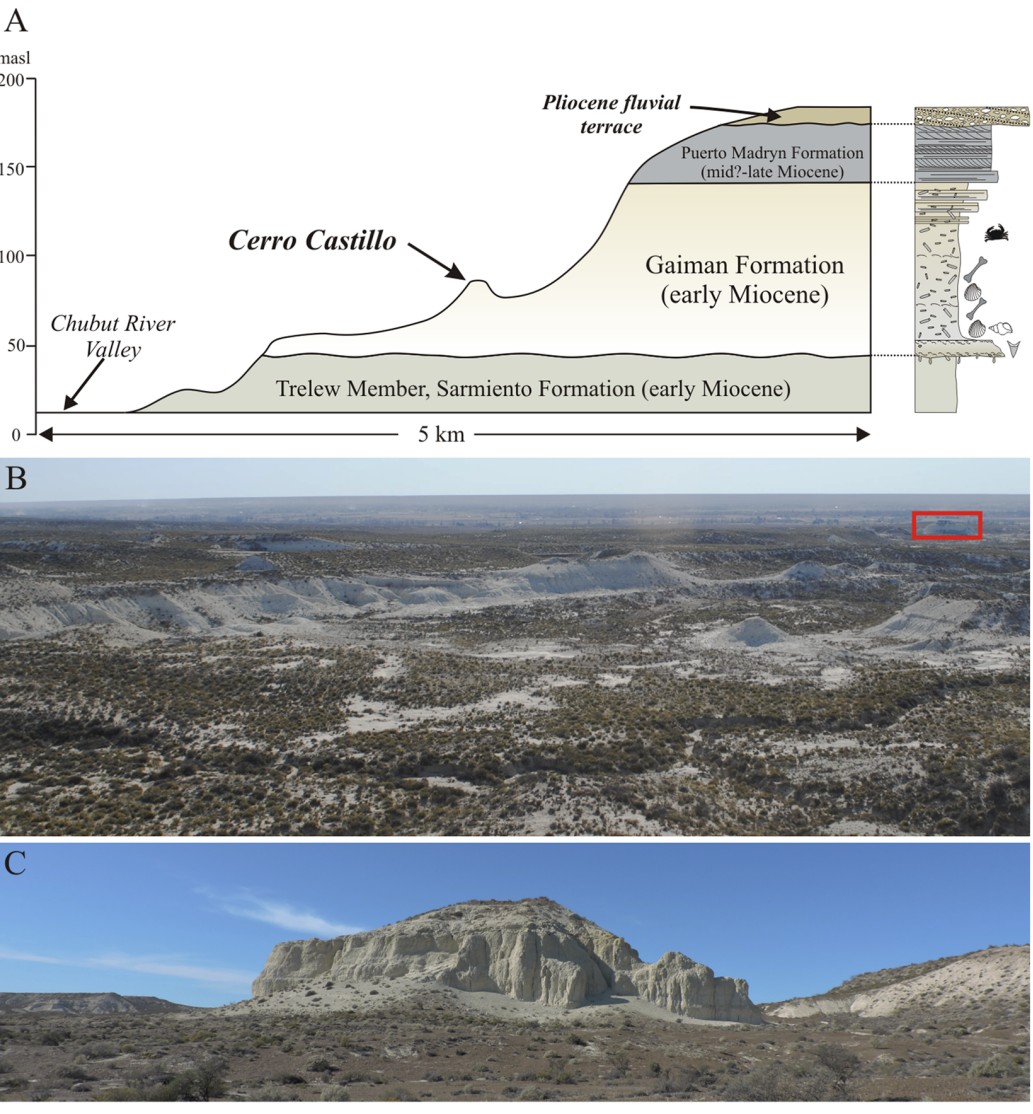

**Figure 3 Stratigraphic section for the type and referred specimens of *Morenocetus parvus*.** (A) Stratigraphic section of Neogene deposits at Cerro Castillo area (south of Trelew city) (indicated in Fig. 2) modified from *Scasso & Castro (1999)*. (B) General view to the northwest of the Gaiman Formation exposure at the Lower Valley of Chubut river; red rectangle indicates the locality of Cerro Castillo. (C) Cerro Castillo.               

(*Fleagle & Bown, 1983*; *Scasso & Bellosi, 2004*). In turn, the Gaiman Formation is conformably overlain by a 30 m thick succession of cross-bedded sandstones and heterolithic deposits referred to the Puerto Madryn Formation (Fig. 3A), which accumulated in estuarine environments.

Unfortunately, *Cabrera (1926)* did not provide precise geographic or stratigraphic information on which horizon the specimen was collected from. However, field inspection of the area south of Trelew city provide insights about the stratigraphic provenance of the specimens. In Cerro Castillo, the basal stratum of the Gaiman Formation is a thin transgressive shell bed with bones and teeth from marine vertebrates

(Fig. 3A; *Cione, 1978*; *Scasso & Castro, 1999*). The marine sediments overlying this basal stratum correspond to the lower half (∼50 m) of the Gaiman Formation. It is composed of white, tuffaceous, thoroughly bioturbated mudstones with occasional thin oyster horizons, deposited in a shallow shelf (*Scasso & Castro, 1999*). In contrast to the upper part of Gaiman Formation, the lower beds produce several disarticulated cetacean remains suggesting that *Morenocetus* was probably collected in this part of the unit.

The age of the Gaiman Formation is based on stratigraphic correlations to other absolutely dated sections in Patagonia and biostratigraphic data. The "Patagoniense" marine deposits, equivalent to the Gaiman Formation, were dated in the Austral Basin (Patagonia, Santa Cruz Province, Argentina) by means of U-Pb and Sr-Sr methods as early Miocene (Aquitanian–Burdigalian; ∼23–18 Ma) (*Parras, Dix & Griffin, 2012*; *Cuitiño et al., 2012*, *2015a*). Equivalent beds in the Comodoro Rivadavia region (Chubut province), dated by the Sr-Sr method, comprise a younger depositional age, spanning from the early to the middle Miocene (19.69 and 15.37 Ma; Burdigalian–early Langhian) (*Cuitiño et al., 2015b*). The initial flooding phase of the Patagoniense transgression occurred in the Aquitanian to early Burdigalian (*Cuitiño et al., 2015b*), whereas the younger deposits of the regressive interval extends only locally to the Langhian stage (*Cuitiño et al., 2015b*). The Cerro Castillo beds analyzed here are part of the lower beds of the Gaiman Formation and can be considered as the initial phase of the "Patagoniense" marine cycle. Based on regional correlations, an early Miocene (Aquitanian–Burdigalian) age is proposed for these cetacean bearing beds. An early Miocene age for the lower part of the Gaiman Formation is also suggested by the Colhuehuapian mammal fauna recovered from the underlying Trelew Member of the Sarmiento Formation (*Flynn & Swisher, 1995*; *Dunn et al., 2013*). The age of the Colhuehuapian fauna was estimated between ca. 21.0 and 20.5 Ma (late Aquitanian) at Gran Barranca (*Dunn et al., 2013*). Based on this information, the age of the overlying Gaiman formation should not be older than late Aquitanian. In addition, evidence from marine vertebrates in the Gaiman Formation (i.e., fishes and penguins) (*Cione et al., 2011*) as well as a palynological assemblage recovered from the study area (*Palazzesi, Barreda & Scasso, 2006*) also indicates an early Miocene age (Burdigalian).

# RESULTS

## Systematic palaeontology

Cetacea Brisson, 1762

Neoceti Fordyce & Muizon, 2001

Mysticeti Gray, 1864 *sensu Cope, 1869*

Chaeomysticeti Mitchell, 1989

Balaenidae Gray, 1825

*Morenocetus Cabrera, 1926*

**Type species by monotypy:** *Morenocetus parvus Cabrera, 1926*.

*Diagnosis:* As for type and only species.

***Morenocetus parvus*** *Cabrera, 1926*

**Holotype:** MLP 5–11, incomplete cranium including the left periotic and incomplete right periotic in articulation with the basicranium, but lacking the rostrum (Figs. 4–11; Tables 2 and 3).

**Referred specimens:** MLP 5–15, cranium including the left periotic (Figs. 5 and 6; Table 2).

**Type locality, horizon and age:** The holotype and referred specimens were collected by Cremonessi at the Cerro Castillo locality, in front of Trelew city, Chubut province, central Patagonia, Argentina; Gaiman Formation (early Miocene, Burdigalian; *Mendía & Bayarsky, 1981*; *Scasso & Castro, 1999*) (Figs. 2 and 3).

**Emended diagnosis:** *Morenocetus* is a small sized balaenid (aproximatly 5.2–5.6 m in TL) which differs from all other Balaenidae in the following unique combination of apomorphies: narrow exposure of the squamosal lateral to the exoccipital (in posterior view the transverse width of squamosal is less than 15% of the distance between sagittal plane and lateral edge of the exoccipital) and dorsal extension of the tensor tympani muscle as a deep canal on the medial side of the anterior process of the periotic.

*Morenocetus* differs from *Peripolocetus* in having a zygomatic process of the squamosal dorsoventrally expanded, in the lack of a distinct ridge delimiting the insertion surface of the tensor tympani muscle on the medial side of the anterior process, and in the lack of a dorsal deflection of the anterodorsal corner of the anterior process of the periotic.

*Morenocetus* differs from *Balaenula* sp., *Balaenula astensis*, *Balaenella*, *Eubalaena* and *Balaena* in having a transversely short supraorbital process of the frontal (the transverse length represents up to twice the anteroposterior length of the supraorbital process), paired tubercles on the supraoccipital limited to low ridges forming the lateral edges of a medial fossa (except *Eubalaena australis* and *Eubalaena japonica*), and straight lateral edges of the supraoccipital.

*Morenocetus* further differs from *Balaenula* sp., *Balaenula astensis* and *Balaenella* in having a postorbital process of the frontal oriented posteriorly, in having a crest-like parieto-squamosal suture, and in having a compound posterior process of the periotic posterolaterally oriented with respect to the longitudinal axis of the anterior process; from *Balaenula* sp. and *Balaenula astensis* in having the anterior edge of the supraorbital process of the frontal pointing posteriorly, presence of supramastoid crest of zygomatic process; from *Balaenella* in having a dorsal margin of the orbit located roughly halfway between the vertex and the ventral surface of the postglenoid process, in the lack of a narial process of the frontal separating the posterior portion of the nasals, in the lack of a tubercle at the junction of the parieto-squamosal and supraoccipital sutures, lack of a lateral tuberosity of the periotic, and in the lack of a ridge delimiting the insertion for tensor tympani muscle.

Differs from *Balaena ricei* and *Balaena mysticetus* in having a short anterior process of periotic, in the lack of a lateral tuberosity on the periotic and a distinct ridge delimiting the insertion surface for tensor tympani muscle on the medial side of the anterior process of periotic, and in the lack of a hypertrophied suprameatal fossa; from

*Balaena myticetus* and *Balaena montalionis* in having a compound posterior process of the periotic exposed on the lateral skull wall; *Balaena mysticetus* in having the anterior edge of the supraorbital process of the frontal pointing posteriorly, optic canal ventrally open, apex of the zygomatic process of squamosal anteroventrally deflected, presence of a supramastoid crest along the dorsal surface of the zygomatic process of squamosal (but does not reach the tip of the zygomatic process), postglenoid process ventrally oriented, the proximal opening of facial canal, internal acoustic meatus and aperture for cochlear aqueduct aligned anteroposteriorly; from *Balaena montalionis* in the lack of a tubercle at the junction of the parieto–squamosal–supraoccipital sutures, and pterygoid sinus fossa located anterior to the foramen pseudovale.

Differs from *Eubalaena* (except *Eubalaena shinshuensis*) in having the supraorbital process of the frontal gradually sloping away ventrolaterally from the skull vertex, posteriorly oriented postorbital process, orbit positioned at half of the vertical distance between the vertex and the ventral surface of the postglenoid process, thicker orbital rim with a flat lateral surface, zygomatic process of squamosal anteroventrally deflected, ventrally oriented postglenoid process, foramen pseudovale not raised above the lateral portions of the squamosal; from *Eubalaena australis*, *Eubalaena japonica* and *Eubalaena glacialis* in having a ventrally open optic canal; a supramastoid crest along the dorsal surface of the zygomatic process of the squamosal; pterygoid sinus fossa located anterior to the foramen pseudovale; in lacking a distinct ridge delimiting the insertion surface of the tensor tympani muscle on the medial side of the anterior process of the periotic; compound posterior process of the periotic posterolaterally oriented with respect to the longitudinal axis of the anterior process, a rounded and short anterior process of the periotic, caudal tympanic process of the periotic with a ventrally oriented ventral border, compound posterior process of periotic exposed in the lateral skull wall, in lacking a hypertrophied suprameatal fossa and lateral tuberosity on the periotic; from *Eubalaena australis*, *Eubalaena japonica*, *Eubalaena glacialis* and *Eubalaena shinshuensis* in the posteriormost point of the exoccipital located more anteriorly than the posterior edge of the occipital condyle.

## Description

**Preservation:** MLP 5–11 is an incomplete skull with a well-preserved cranium, lacking: the rostrum; the right supraorbital process of the frontal (broken 180 mm from the sagittal plane of the skull); a small posterodorsal portion of the supraoccipital anterior to the foramen magnum; both jugals and palatines; the anterior portion of the vomer; the right alisphenoid; much of the presphenoid and mesethmoid; most of the right postglenoid process and glenoid fossa; the apex of the zygomatic process of the squamosal; both tympanic bullae; and most of the right periotic. Most of the ventral surface of the cranium is eroded and broken. However, both periotics are preserved articulated with the skull. The specimen MLP 5–15 shows a similar state of preservation to the holotype, with the difference that the zygomatic and postglenoid processes of the left squamosal are complete.
**Physical maturity:** In MLP 5–11 and MLP 5–15 the basioccipital/exoccipital and supraoccipital/exoccipital sutures are completely fused (SR4). In *Eubalaena australis*, the supraoccipital/exoccipital suture (the last occipital suture to fuse during ontogeny in Balaenopteroidea; *Walsh & Berta, 2011*) is already completely fused in subadults (Table 1). In MLP 5–11 and MLP 5–15 fractures in the ventral surface of the median basicranium approximate the position of the basioccipital/basisphenoid suture, which is closed (SR3; the suture is clearly obliterated in the floor of the cranial cavity). In both MLP 5–11 and MLP 5–15 the basisphenoid/presphenoid suture is wide open. Other sutures of the cranium of the holotype specimen (pterygoid/vomer/squamosal/ alisphenoid-squamosal/pterygoid/parietal) are closed, but visible (SR3). Additional adult features of the *Morenocetus* crania include: occipital condyles with smooth external surfaces (punctate in juvenile mysticetes); compact bone forming the external surface of cranial elements; and a sharp, salient nuchal crest. The latter features are characteristic of demonstrably adult specimens of Balaenidae, e.g., USNM 267612 (*Eubalaena australis*), USNM 23077 (*Eubalaena glacialis*) and USNM 257513 (*Balaena mysticetus*). Together, these observations suggest that MLP 5–11 and 5–15 are not juveniles and probably represent at least subadult, but not physically mature individuals.

**Body size:** Based on the equation of *Pyenson & Sponberg (2011)*, a conservative estimate for the total body length of *Morenocetus* is 5.2 m (based on BZW of the holotype specimen). In MLP 5–15 the left zygomatic process is completely preserved, which allows an estimated BZW of ∼620 mm. In this case, the estimated TL is 5.6 m. This value is slightly lower than those of late Miocene and Pliocene balaenids such as: *Balaenula astensis* 6.6 m; *Balaena montalionis* 7.2 m; *Balaenella brachyrhynus* 7.9 m; *Eubalaena shinshuensis* 12.5 m and *Eubalaena ianitrix* 13.8 m.

On the other hand, the estimated total body length of *Morenocetus* based on the regression equation of *Lambert et al. (2010)* is 5.34 m for the holotype and 5.64 m for MLP 5–15. For the other extinct balaenids the values obtained are: *Balaenula astensis* 6.41 m; *Balaena montalionis* 7.39 m; *Balaenella brachyrhynus* 8.05 m; *Eubalaena shinshuensis* 12.89 m and *Eubalaena ianitrix* 14.29 m. These values are similar to those obtained with the *Pyenson & Sponberg (2011)* equation. In *Balaenella* the TL could be underestimated because both zygomatic processes are incomplete.

**General shape of the skull:** The skull of the holotype and MLP 5–15 are comparable in size. The BZW is 570 + mm in MLP 5–11, which corresponds to ∼25 % of the BZW of extant physically mature balaenids. In dorsal view, the occipital shield is anteroposteriorly elongated and triangular-shaped with the anterior margin rounded. It extends to a point posterior to the level (anteroposteriorly) of the anterior edge of the orbit. The orbits open dorso-laterally and are positioned high on the skull. The supraorbital process of the frontal, gradually sloping from the vertex of the skull, is transversely short and anteroposteriorly broad and it is mainly transversely oriented with respect to the sagittal axis of the skull in dorsal view. The anterodorsal portion of the parietal overlaps the posterodorsal region of the frontal, extending forward until the level of the anterior

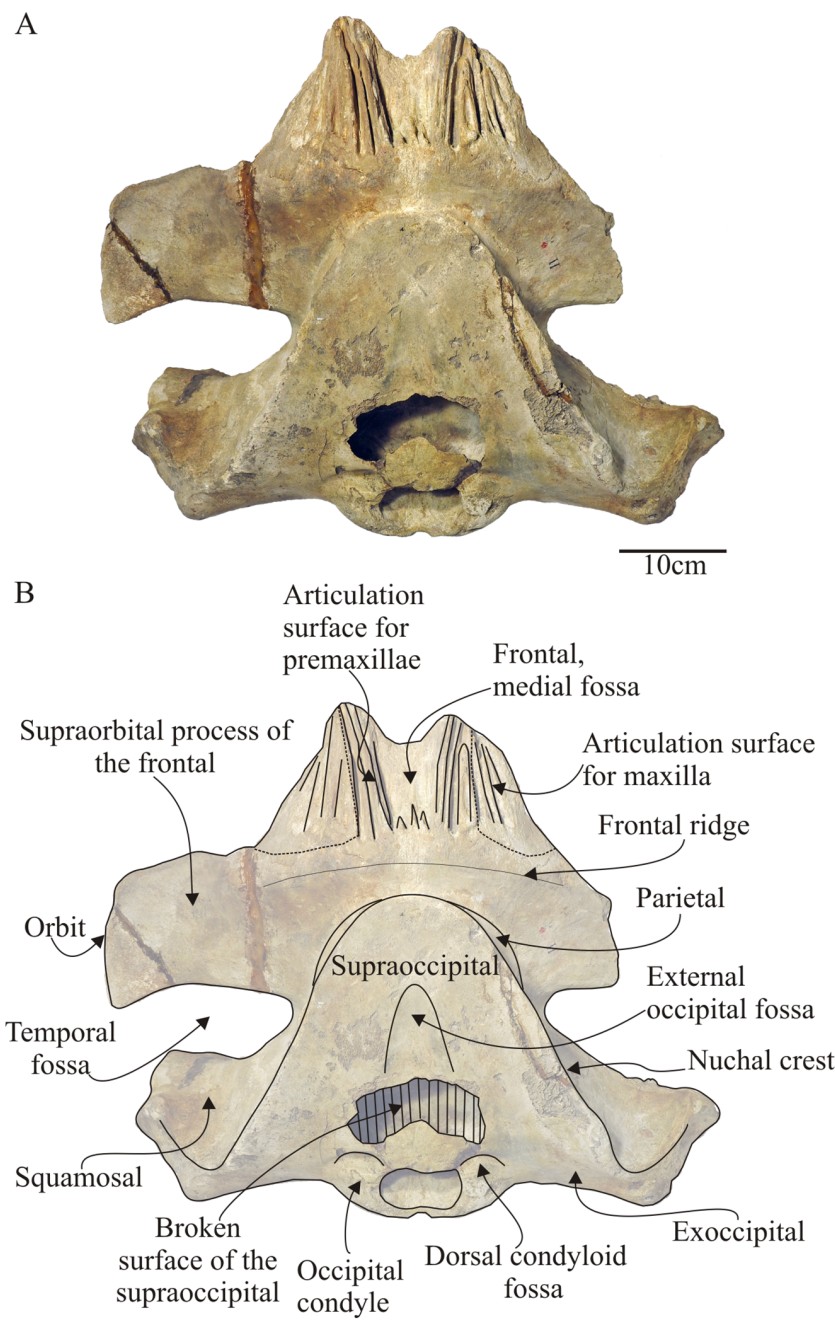

**Figure 4** *Morenocetus parvus*, **holotype, MLP 5–11, cranium.** (A) Dorsal view. (B) Key features in dorsal view. Hatching indicates major breaks.

margin of the occipital shield. The zygomatic process is short, anterolaterally oriented and ventrally deflected.

**Supraoccipital:** In dorsal view (Fig. 4), the supraoccipital does not reach the level of the preorbital process of the supraorbital process of the frontal. It has a triangular outline with a rounded anterior apex. Its lateral margins are almost straight in dorsal view forming a ~45° angle; the lambdoid suture with the parietal and squamosal is exposed in

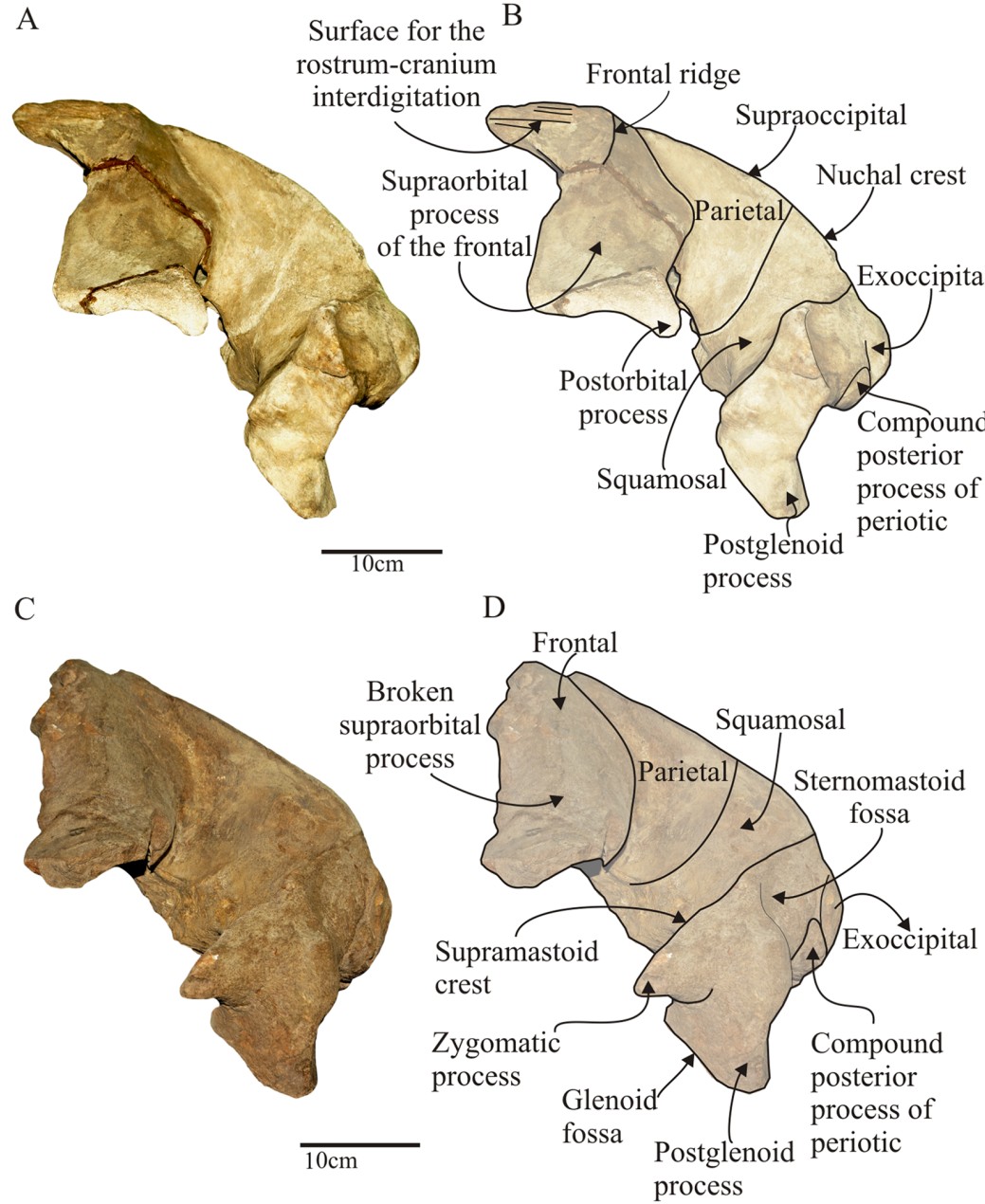

**Figure 5 _Morenocetus parvus_, crania.** (A) Holotype, MLP 5–11, cranium in lateral view. (B) Key features in lateral view. (C) Referred specimen, MLP 5–15, lateral view. (D) Key features in lateral view.

dorsal and lateral views. The lateral margins of the supraoccipital form a pronounced nuchal crest. The anterior margin of the supraoccipital contacts with the frontal through an observable joint (state SR3) and is not raised as in _Balaenella_ (_Bisconti, 2005_). The dorsomedial surface of the supraoccipital bears a low, triangular and anteroposteriorly oriented (length = 90 mm) depression that does not reach the anterior margin of the supraoccipital (extending to 75 mm from this margin)—here termed the external occipital fossa (new term). There is no external occipital crest. In lateral view (Fig. 5) the

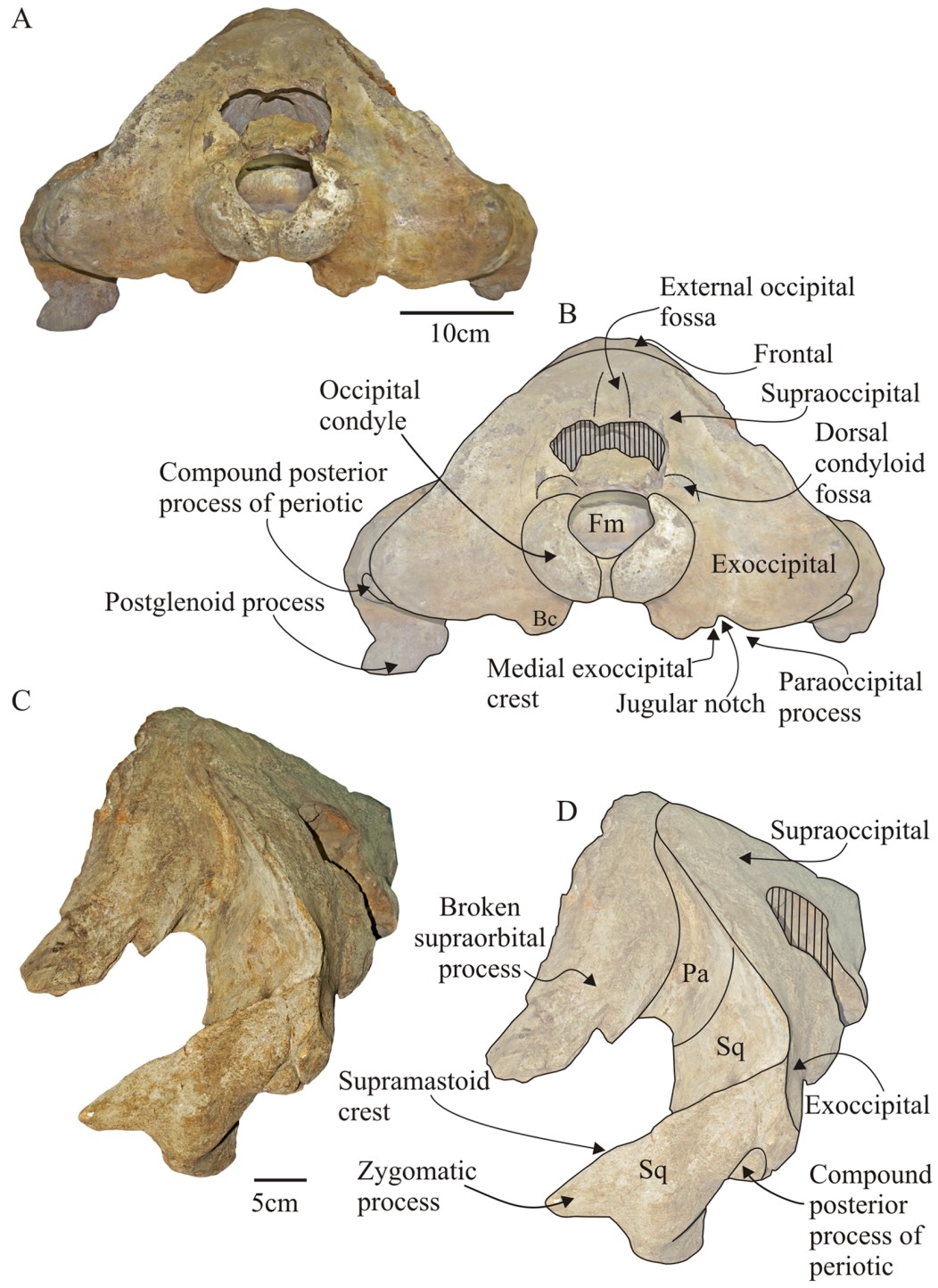

**Figure 6 *Morenocetus parvus*, crania.** (A) Holotype, MLP 5–11, posterior view. (B) Key features in posterior view. (C) Referred specimen, MLP 5–15, posterolateral view (D) Key features in left posterolateral view. Abbreviations: Bc, basioccipital crest; Fm, foramen magnum; Pa, parietal; Sp, squamosal. Hatching indicates major breaks.

**Table 2 Skull measurements of the holotype (MLP 5–11) and referred specimens (MLP 5–15) of *Morenocetus parvus* in mm.**

| Measurement | MLP 5–11 | MLP 5–15 |
|---|---|---|
| Bizygomatic width | +570 | +530 |
| Width of the skull at the level of the exoccipitals | 470 | +385 |
| Supraoccipital width anterior to foramen magnum | 320 | +340 |
| Supraoccipital width at mid-lenght | 220 | 220 |
| Supraoccipital width at 10 cm to the anterior margin | 195 | 185 |
| Supraoccipital length | 270 | 280 |
| Transverse diameter of left occipital condyle | 65 | 75 |
| Idem right | 65 | 75 |
| Dorsoventral diameter of left occipital condyle | 130 | 140 |
| Idem right | 130 | 145 |
| Transverse diameter of foramen magnum | 75 | 70 |
| Distance between lateral border of left occipital condyle and lateral border of exoccipital | 155 | 130 |
| Width of the occipital condyles plus foramen magnum | 155 | 200 |
| Anteroposterior diameter of the supraorbital process of the frontal in the distal edge | 132 | – |
| Transverse diameter of the supraorbital process of the frontal | 345 | 320 |
| Anteroposterior diameter of the supraorbital process of the frontal in the constriction | 122 | – |
| Width of the optic canal in its medial portion | 36 | – |
| Width of the optic canal in its lateral portion | 92 | – |
| Length of the optic canal | 275 | – |
| Orbital length (between preorbital and postorbital process) | 130 | – |
| Length of the interorbital region | 52 | 50 |
| Length of the optic canal | 275 | – |
| Anteroposterior diameter of the temporal fossa | 90 | 100 |
| Transverse diameter of the temporal fossa | 190 | 210 |

supraoccipital is dorsally convex and obliquely oriented forming an acute angle with the sagittal plane of the skull. In posterior view (Figs. 6A and 6B) the supraoccipital is not extensively exposed above the dorsal margin of the foramen magnum: in *Morenocetus* the posterior aspect of the supraoccipital is approximately twice the height of the foramen magnum, while in *Eubalaena* its height is about four-fold that of the foramen magnum (e.g., USNM 267612, CNPMAMM 774). The suture with the exoccipital is not observable (stage SR4).

**Exoccipital:** The exoccipital is completely fused anteriorly to the supraoccipital via a closed suture (SR4) (Figs. 6A and 6B). Ventrolaterally it contacts the squamosal through a distinct occipitosquamosal suture (stage SR3). This suture begins at the lateral margin of the exoccipital, where its outline is convex, and then curves ventrally continuing inside the periotic fossa where it becomes indistinct.

In dorsal view (Fig. 4) the posterior surface of the exoccipital is slightly concave in the sagittal plane and the posterior margins are laterally oriented forming an almost right angle with the sagittal plane of the skull. In posterior view (Fig. 6A), the exoccipital is transversely wide (representing 84% of the BZW) forming most of the occipital shield.

The occipital condyles form the ventral–ventrolateral margins of the foramen magnum and are large in comparison with the size of the cranium: the maximum width of the occipital condyles represents 28% of the BZW (Table 2) (unlike extant balaenids where it represents ~15%). Both occipital condyles are well preserved, ovoid-shaped and in dorsal view are not as convex in the sagittal plane as in extant balaenids (e.g., *Eubalaena australis* USNM 267612; *Eubalaena glacialis* USNM 23077 and *Balaena mysticetus* USNM 257513). In lateral view, the transverse axis of the occipital condyles is oriented forming an acute angle with the horizontal plane of the skull. The occipital condyles are separated in the ventromedial portion by a shallow and narrow intercondylar notch (length = 60 mm; minimal width = 4 mm). The foramen magnum has an almost circular outline (75 × 75 mm). Posterior to the nuchal crest in the dorsolateral margin of each exoccipital there is a well-defined transversely oriented groove. Anterodorsal to the occipital condyle there is a distinct dorsal condyloid fossa. There is no distinct ventral condyloid fossa. The lateral margin of the exoccipital is convex and the ventral margin forms a deep jugular notch and short paroccipital process. Medial to the jugular notch, there is a small and rounded ventral projection, the medial exoccipital crest (Fig. 6A), similar to that observed in extant balaenids (*Eubalaena australis* USNM 267612, *Eubalaena glacialis* USNM 23077 and *Balaena mysticetus* USNM 257513), as well as in the toothed mysticete *Aetiocetus weltoni* (*Deméré & Berta, 2008*). The small paroccipital process is located lateral to the jugular notch and dorsal to the basioccipital crest. The paroccipital process is triangular, dorsoventrally short, and its posterior surface is flat. In ventral view (Fig. 7), the compound posterior process of the tympanoperiotic contacts the exoccipital through a visible (SR3), straight and posterolaterally oriented suture. The suture between the exoccipital and basioccipital is obliterated (SR4). The contribution of the exoccipital to the formation of the basioccipital crest could not be established due to the advanced fusion of the exoccipital–basioccipital suture. The paroccipital process is anterior to the occipital condyles and its ventromedial surface is smooth and triangular marking the presence of the posterior sinus fossa. The articulation of the stylohyal could not be determined because the ventral surface of the paroccipital process is eroded. Medial to the paroccipital process there is a deep and anteroposteriorly oriented sulcus that corresponds to the path of the hypoglossal (XII), glossopharyngeal (IX), vagus (X) and spinal (XI) nerves and the jugular vein and the external carotid artery that exit the cranium through the jugular notch. The posterior margin of the exoccipital extends anterior to the occipital condyles.

**Basioccipital:** The basioccipital is divided into a medial horizontal portion or basilar part, forming the floor of the cranium, and two ventrolateral projections forming the prominent basioccipital crests (Fig. 7). The basilar part is slightly concave in posterior view and its anterior contact with the basisphenoid is hidden by the posterior extension of the vomer. However, in the dorsal surface of the basioccipital a rugose surface of bone is visible that corresponds to the basisphenoid–basioccipital suture. In MLP 5–15 the vomer is absent, thus the basisphenoid–basioccipital suture is visible in ventral view (SR3 stage). In posterior view (Fig. 6), the basioccipital crest is bulbous being mediolaterally and

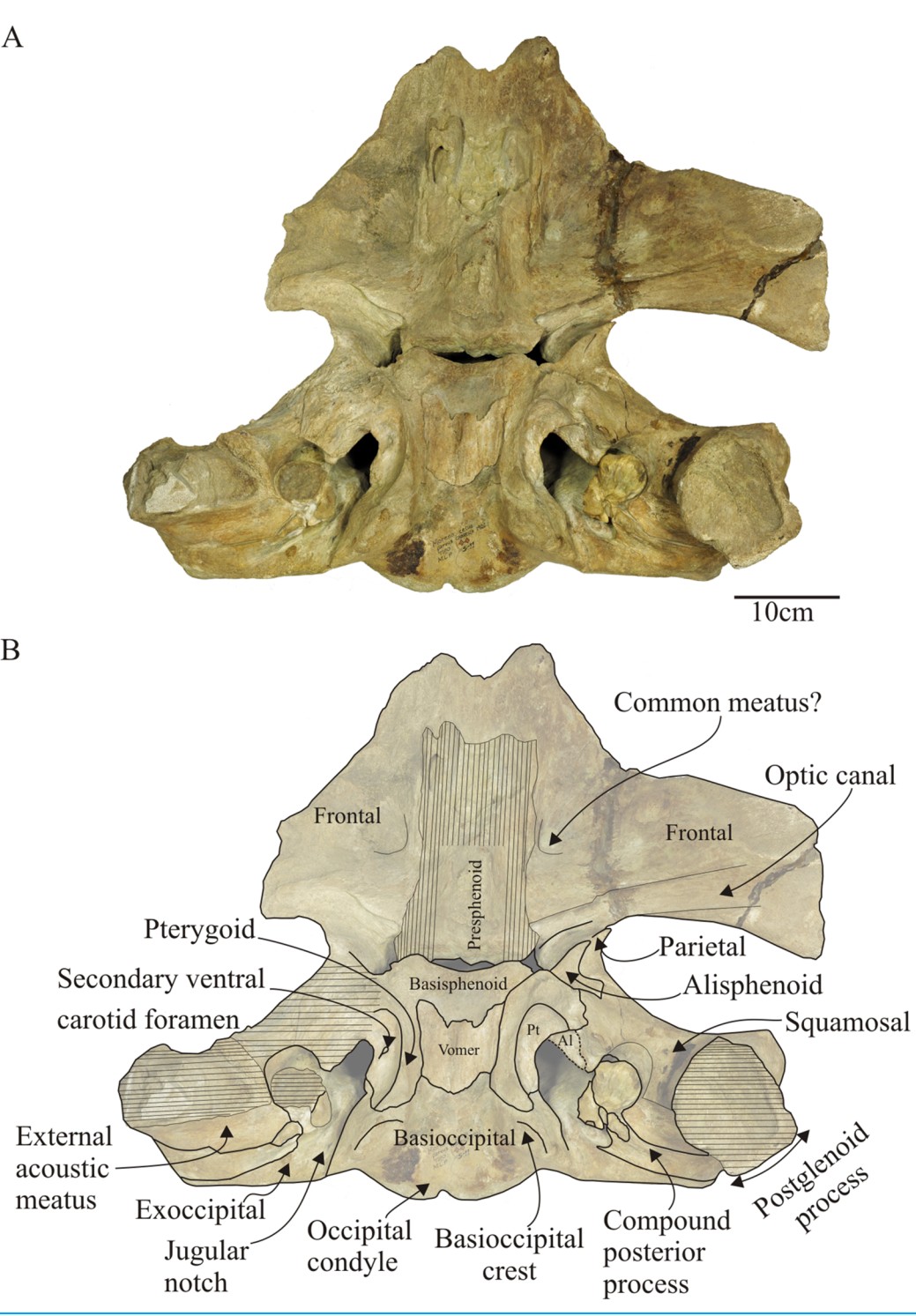

**Figure 7 *Morenocetus parvus*, holotype, MLP 5–11, cranium.** (A) Ventral view. (B) Key features in ventral view. Al, alisphenoid; Pt, pterygoid. Hatching indicates major breaks.

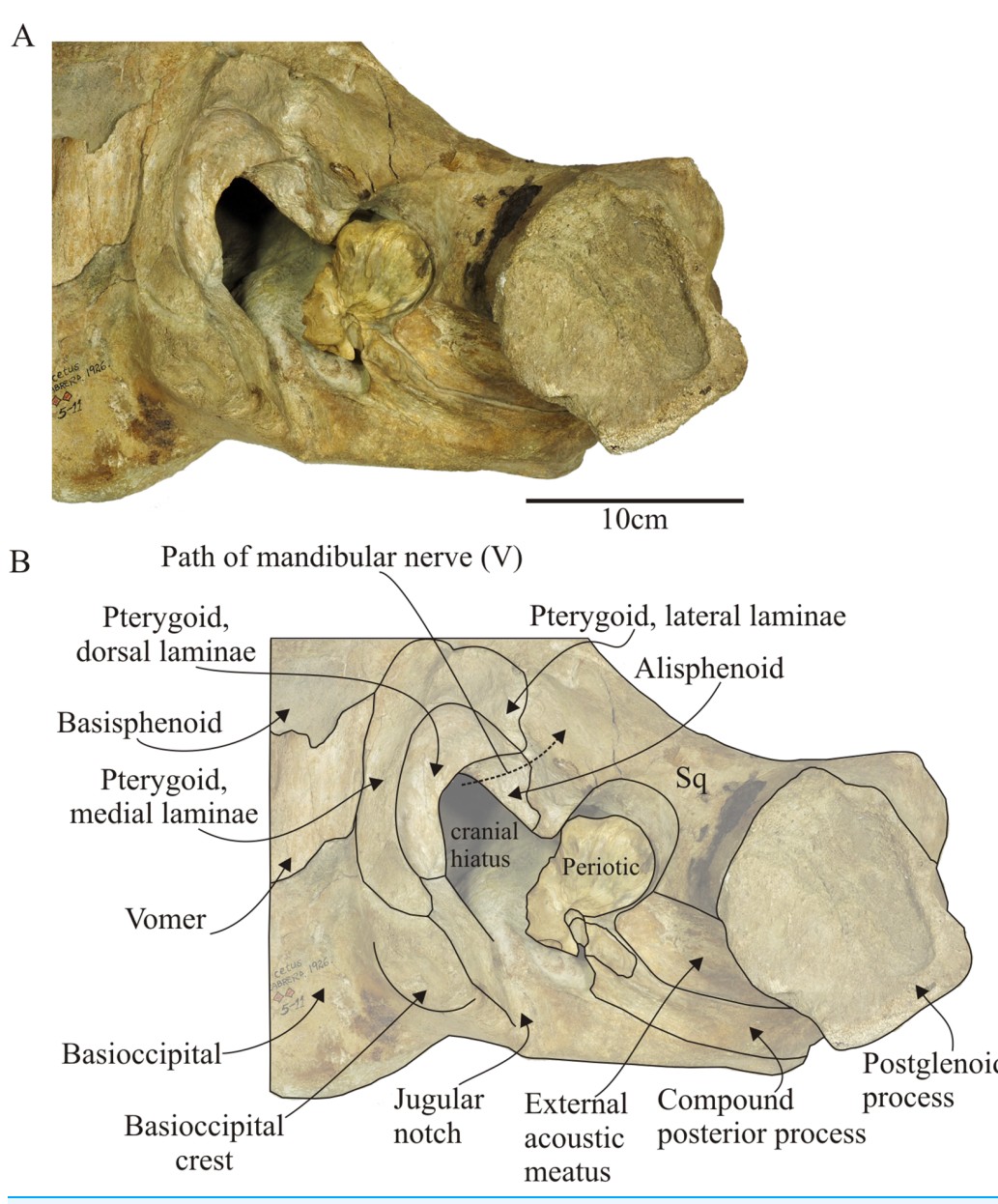

Figure 8 *Morenocetus parvus,* holotype, MLP 5–11, basicranium. (A) Ventral view showing the articulated left periotic. (B) Key features in ventral view. Sq, squamosal.

dorsoventrally thickened. The medial surface of the process is anteriorly convex and posteriorly flat; the ventral margin has a rugose texture suggesting the presence of cartilage. In MLP 5–15 the ventro-medial surface of the basioccipital crest presents ridges and furrows, which may represent for the site of origin of the scalenus medius muscle (*Schulte & Smith, 1918*: Fig. 16, p. 49). The basioccipital crests contact anteriorly with the middle pterygoid lamina through a closed but observable suture (stage SR3).

**Basisphenoid–Presphenoid:** Posteriorly, the basisphenoid contacts the basioccipital, laterally with the middle pterygoid lamina and anteriorly with the presphenoid (Figs. 7 and 8).

The dorsal carotid foramen is not observable in the dorsal surface of the basisphenoid. The presphenoid is elongated anteroposteriorly and contacts laterally with the frontal and posteriorly with the basisphenoid through an open basisphenoid-presphenoid fissure (width = 73 mm; length = 7 mm). The suture with the orbitosphenoid is indistinct. Most of the ventral surface of the presphenoid is eroded; the preserved lateral portions of the presphenoid suggest a triangular shape in ventral view as in other balaenids. Its posterolateral portion contributes to the delimitation of the orbital fissure and the well-defined notch for the passage of the optic nerve.

**Vomer:** Only the posterior portion of the vomer is preserved. As the palatines and pterygoid hamulus are not preserved, the posterior part of the vomer is exposed on the ventral wall of the cranium. The preserved portion of the vomer corresponds to the dorsal and dorsomedial wall of the internal nasal passage (Figs. 7 and 8). It is horizontally orientated with respect to the sagittal plane of the skull and placed ventral to the basisphenoid and basioccipital underlapping the basioccipital/basisphenoid suture. The vomer extends from the level of the anterior margin of the pterygoid sinus fossa until the anterior 2/3 of the medial lamina of the pterygoid. However, the posterior margin is not preserved, therefore the posterior extent could not be determined. It is wide anteriorly and becomes narrower posteriorly. The lateral edges of the vomer are sharp. The median vomerine crest does not reach the preserved posterior margin of the vomer.

**Pterygoid:** The preserved portions of the pterygoid correspond to the medial, and dorsal laminae and part of the lateral (Fig. 8) and a small portion of the ventral lamina in MLP 5–15. The pterygoid hamuli are not preserved; therefore the medial and dorsal laminae of the pterygoid are exposed in the ventral surface of the skull. The medial lamina is anteroposteriorly oriented and medially contacts the basioccipital and basisphenoid for ~80 mm of its length, anteriorly with the basisphenoid and posteriorly with the medial and anterolateral surface of the basioccipital crest. The dorsal lamina extends laterally and contacts the squamosal through a visible and irregular suture (length of the suture along which the pterygoid and squamosal contact ~40 mm), and posteriorly where it contacts with the basioccipital. In *Morenocetus* the dorsal lamina of the pterygoid is posteriorly elongated, resulting in the pterygoid sinus fossa being anteroposteriorly longer than it is dorsoventrally deep. This contrasts with the condition in extant balaenids (e.g., *Eubalaena australis*: CNPMAMM 774, NMNZ 2239), where the pterygoid sinus fossa is principally developed dorsoventrally. One-third of the pterygoid sinus fossa is located anterior to the level of the foramen pseudovale. A well-defined foramen followed by a short posterolateral fissure is identified on the dorsal lamina of the pterygoid, ~51 mm from the pterygoid–basioccipital suture. The same structures are present in the medial lamina of the pterygoid of neonates (e.g., USNM 500860, CNPMMAM 746) and adult specimens (CNPMAMM 774, NMNZ 2239) of *Eubalaena* spp. Observations in neonate specimens shows that this fissure connects with the basisphenoid, and therefore could be homologized with the path of the internal carotid artery into the cranial cavity (R. E. Fordyce, 2016, personal communication). The strict ventral carotid foramen in the basisphenoid is not present in neonate specimens of *Eubalaena* spp., and the foramen in

the pterygoid is provisionally referred to here as the "secondary ventral carotid foramen" (Fig. 7B). The dorsal opening of the internal carotid artery is nonpatent in the dorsal surface of the basisphenoid. The pharyngeal crest, which separates the pharyngeal and ear regions of the skull, is formed anteriorly by the medial lamina of the pterygoid and is continuous posteriorly with the basioccipital crest; this crest is dorsoventrally lower than in *Eubalaena* (e.g., CNPMAMM 774, NMNZ 2239).

**Alisphenoid:** In ventral view (Figs. 7 and 8), the alisphenoid contacts along its dorsal surface with the parietal and ventrally it is overlapped in its posteromedial portion by the dorsal lamina of the pterygoid and posterolaterally by the squamosal through well-defined sutures. The alisphenoid is small with a maximum with of ~41 mm and a length of ~56 mm. It presents a well-defined lateral and triangular-shaped projection that extends ~28 mm over the parietal that does not reach the lateral margin of the skull; therefore it is not exposed in the medial wall of the temporal fossa. Most of the ventral surface of the alisphenoid is eroded. The lateral projection of the alisphenoid delimits the posteroventral margin of the optic canal. The alisphenoid contacts anteriorly with the parietal through a visible suture that extends mediolaterally. Posterolaterally, the alisphenoid contacts the squamosal through a convex suture and posteromedially with the pterygoid. The foramen pseudovale does not perforate the alisphenoid externally, but is instead confined to the squamosal (Fig. 8).

**Squamosal:** The squamosal is a massive bone primarily developed in the sagittal plane and also forming the short and anterolaterally oriented zygomatic process.

In lateral view (Fig. 5), the squamosal is clearly elongated dorsoventrally. In posterior view the squamosal–exoccipital suture has a laterally convex profile. In MLP 5–15 a robust laterally directed supramastoid crest forms the dorsal surface of the zygomatic process of the squamosal (Figs. 5D, 6C and 6D). A small squamosal prominence is present. The zygomatic process is short, has a rounded apex, is anterolaterally oriented and ventrally deflected (Fig. 5C and 5D). Based on the combined morphology of holotype and referred specimen we infer that the apex of the zygomatic process of the squamosal is placed posterior to the postorbital process, but not apposed to it. The postglenoid process is better preserved in MLP 5–15; it is dorsoventrally high, posteroventrally directed in lateral view (Fig. 5D), and ventrally oriented in posterior view. In dorsal view (Fig. 4B), the squamosal forms the posterolateral wall of the temporal fossa. The temporal fossa has an approximately triangular shape. There is no squamosal cleft. The glenoid fossa is positioned anteriorly to the level of the posterior apex of the nuchal crest (Fig. 5D). In ventral view (Fig. 8), the squamosal forms the anterior, dorsal and lateral walls of the periotic fossa. This fossa is deep, with a transverse diameter extending from the base of the falciform process to the anteromedial margin of the external auditory meatus. The external auditory meatus is a deep excavation located in the posterolateral margin of the skull, dorsal to the postglenoid process. The posterior margin is almost straight and defined by the posterior meatal crest that makes contact along a straight and observable suture (state SR3) with the compound posterior process of the tympanoperiotic. The medial apex of the external auditory meatus forms the pointed spiny process of the

squamosal, which extends anteromedially to overhang the lateral margin of the periotic fossa. The ventral surface of the external auditory meatus is concave, and the surface of its medial half is deep with smooth bone texture correlated with the position in vivo of the sac-shaped tympanic membrane or "glove finger". The falciform process is better preserved in MLP 5–15; it is robust and posteromedially oriented. Medial to the falciform process there is a broad and deep notch (width = 12 mm) (=foramen pseudovale) posteriorly oriented (as is observed in extant balaenids) for the passage of the mandibular branch of the trigeminal nerve (V3). The glenoid fossa is almost flat to slightly concave and broad, and anteromedially defined by a distinct ridge as in extant balaenids (e.g., *Eubalaena australis* CNPMAMM 748, MLP 1508). The high position of the orbits with respect to the postglenoid process (viewed laterally) indicates that the position of the glenoid fossa must be posterior to the postorbital process of the frontal.

**Parietal:** In dorsal view (Fig. 4), the parietals are not in contact on the midline of the vertex of the skull, but due to their anteriorly extended position a small portion is visible lateral to the cranial vertex. In lateral view (Fig. 5), the parietal forms part of the lateral skull wall and contacts the frontal anteriorly forming the coronal suture. The latter begins at the level of the anterior apex of the supraoccipital, continuing posteriorly above the posterodorsal margin of the frontal, then bends ventrally and ends at a well-developed notch. Posteriorly, the parietal contacts the squamosal via a parieto-squamosal suture that runs dorsoventrally in the medial wall of the temporal fossa. The parieto-squamosal suture has a posteriorly convex outline and forms a salient crest. Dorsally, the parietal contacts the supraoccipital and contributes to the laterally protruding nuchal crest. The anterodorsal portion of the parietal projects on the posteromedial corner of the supraorbital process of the frontal. In this region the external surface of the parietal is strongly concave (transversely) and smooth.

In ventral view (Fig. 7), the parietal has a small exposure in the posteromedial portion of the optic canal where it contacts anteriorly with the frontal and posteriorly with the squamosal. In this region, the parietal is partly overlapped by the alisphenoid and forms a well-defined notch anteriorly, and a pointed lateral projection posteriorly.

**Frontal:** The frontals form the interorbital region, the supraorbital process of the frontal and the orbital wall. In dorsal view (Fig. 4), exposure of the frontals in the vertex is clearly observed extending anteroposteriorly for 52 mm. The anterior portion of the frontal is exposed due to the disarticulation of nasals, maxillae and premaxillae, exposing the ridges and grooves that characterize the rostrum–cranium interdigitation. In the anterodorsal surface of the frontal, there is a medial fossa for the reception of nasal bones, laterally bounded by a complex of longitudinal ridges that form the articulation surfaces for premaxillae and maxillae (Figs. 9A and 9B). The sutural surface of the frontal for the maxilla suggest the presence of a broadly triangular ascending process of the maxilla (a condition shared with extant balaenids; *Marx et al., 2013*) (Fig. 9D). Posterior to this region, the dorsal exposure of the frontal presents a rounded anterior margin with a low but distinct ridge (Figs. 9B and 9D), also observed in *Eubalaena* spp. which does not correspond to the posteriormost extension of the fronto-maxilla suture. This ridge, here

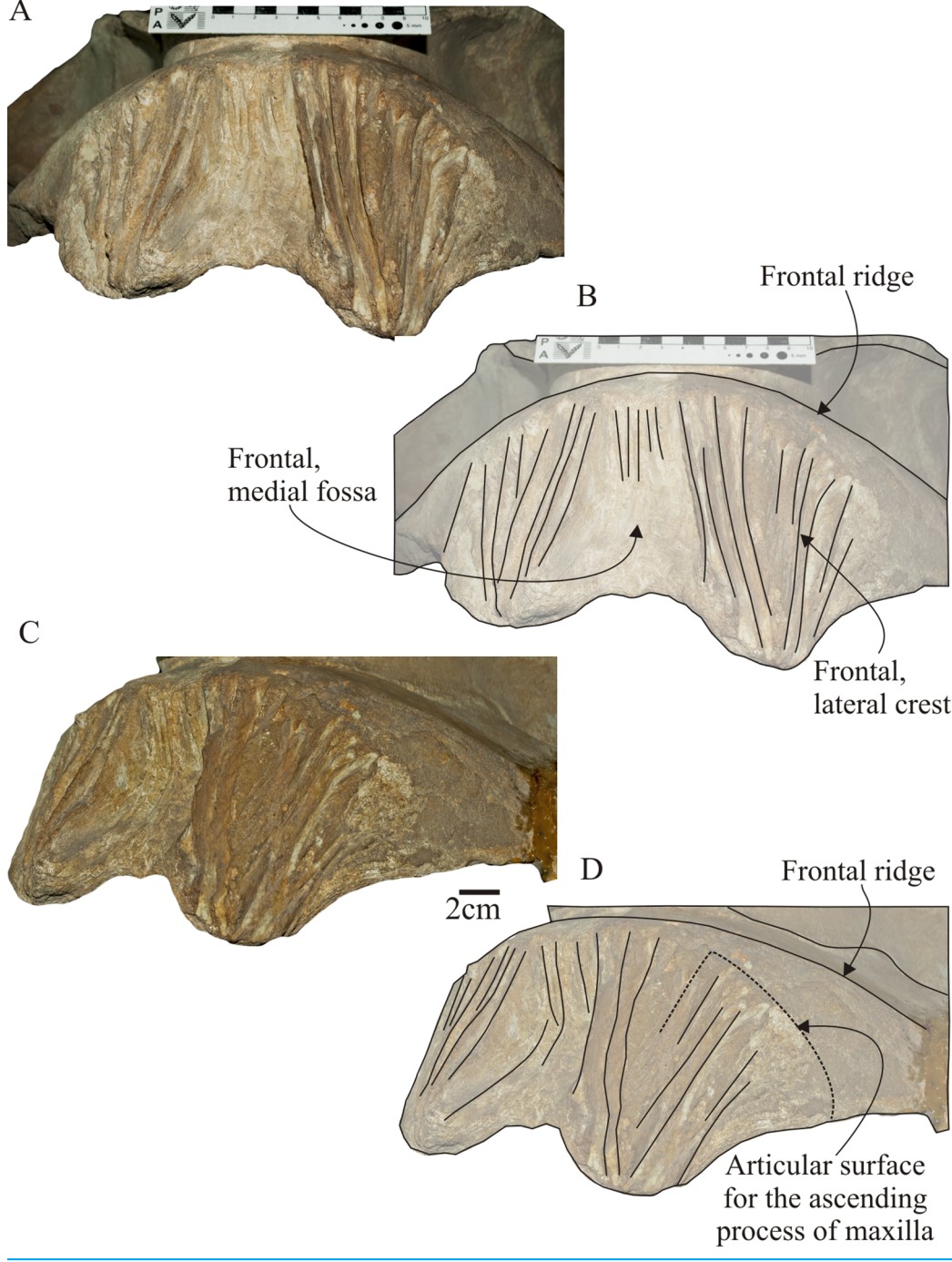

**Figure 9** *Morenocetus parvus*, **holotype, MLP 5–11, cranium.** (A) Anterior view. (B) Key features in anterior view. (C) Left anterolateral view. (D) Key features in left anterolateral view.

named frontal ridge (new term), is not considered as an anterior extension of the orbitotemporal crest, as is observed in other mysticetes (e.g., *Parietobalaena* and *Pelocetus*). Dissection of the temporalis muscle in *Eubalena australis* (CNPMAMM 748; Fig. S1) show that this muscle extends anteriorly until the anteriormost extension of the

fronto-parietal suture, but never reaches the frontal ridge. In addition, dissection of facial and nasal muscles does not show that any of these muscles are attached with this ridge (*Buono et al., 2015*). In the region of contact between the right and left frontal the surface of the bone is rugose, which could correspond to the interfrontalis suture. The frontal gradually slopes ventrolaterally from the vertex of the skull forming the supraorbital process. It is principally developed in the transverse plane. The supraorbital process is longer transversely than anteroposteriorly: the transverse width of the supraorbital process is about twice its anteroposterior length (Table 2). The anterior margin is straight and forms an acute angle (67°) with the sagittal plane of the skull, while its posterior margin is slightly concave. The dorsal surface of the process is roughly flat. In lateral view (Fig. 5), the dorsal margin of the orbit opens laterally at a level dorsal to that of the zygomatic process of the squamosal; the vertical distance between the dorsal margin of the orbit and the ventral margin of the postglenoid process is 185 mm. The antero-posterior length of the orbit (measured between the apex of the preorbital and postorbital process) compared with BZW is 23% (OL/BW), close to the toothed archaic mysticete *Aetiocetus weltoni* (27%, *Deméré & Berta, 2008*: 323), but greater than in extant balaenids such as *Eubalaena australis* and *Eubalaena glacialis* (mean value of OL/BW = 9%, $N = 10$; Table S2). The dorsal margin of the orbit is thickened and formed anteriorly by the incomplete preorbital process and posteriorly by the postorbital process. The postorbital process is robust and posteriorly directed in dorsal view, with a rounded outline. In lateral view, the anteroposterior axis of the orbit is parallel to the horizontal plane of the skull. The ventral margin of the orbit is not preserved due to the lack of the maxilla and jugal. The orbitotemporal crest is present on the anterodorsal surface of the supraorbital process of the frontal but it is poorly developed.

In dorsal view, the supraorbital margin of the orbit is roughly straight while it is dorsally concave in lateral view. The external surface of the frontal is smooth in the interorbital region and in the dorsal surface of the supraorbital process, however in the supraorbital margin of the orbit the bone is rugose/porous indicating the presence of cartilage, as has been observed in calves of *Eubalaena australis* (CNPMAMM 748 and other specimens dissected by M.R.B.). In ventral view (Fig. 7), the optic foramen, the exit of the optic cranial nerve (II), is a notch that opens lateral to the basisphenoid–presphenoid fissure. From that point, the optic canal, the sulcus for the transmission of the optic nerve (II), extraocular muscles and blood vessels, extends laterally along the posteroventral margin of the supraorbital process. The optic canal of *Morenocetus* does not occupy the entire anteroposterior length of the ventral surface of the supraorbital process. Instead, it is developed in the posterior two thirds of the supraorbital process (~70% of the anteroposterior length of the process). It is bounded posteromedially by the parietal. The optic canal is deep and narrow in its medial portion (width = 35 mm) and widens laterally (width = 50 mm). The anterior margin of the optic canal is completely preserved in its lateral half; therefore we infer that the canal remained open ventrally along its entire length. The dorsal wall of the optic canal is pierced by a large single ethmoidal foramen for the exit of the ethmoidal nerve and blood vessels, and by small frontal foramina. The presence of the orbitosphenoid in the optic canal is difficult to establish

due to tightly closed sutures of this element with the frontal and presphenoid. The ventral surface of the frontal presents two well defined and symmetrical fossae placed in the contact between the presphenoid and the frontal, and dorsal to the level of the supraorbital process of the frontal, which might correspond to the common meatus.

**Periotic:** Both periotic bones are preserved in situ sutured against the squamosal. The left periotic is completely preserved unlike the right one which is broken at the anterior and lateral tuberosity. In ventral view (Fig. 10), the anterior process is shorter than the anteroposterior length of the pars coclearis, and presents a rounded anterior apex (Table 3). The small anterior pedicle for the articulation of the tympanic bulla is located on the ventral surface of the anterior process, and it is defined by a sharp crest flanked by two furrows. Lateral to the anterior process, the body of the periotic, lateral to the pars cochlearis is hypertrophied (maximum transverse width of the body of periotic is 62% larger than the maximum width of pars cochlearis, Table 3), and develops ventrolaterally with a rounded lateral margin. There is no distinct lateral tuberosity. Posteromedially, in the body of the periotic, the mallear fossa is represented by a poorly defined concavity. The site of attachment of the tensor tympani muscle is inferred to be a slight depression medial to the anterior process, with a dorsal extension of the muscle attachment on the medial side of the anterior process via a distinct ridge (Fig. 10B). The pars cochlearis is anteroposteriorly oriented, with the lateral margin forming an acute angle with the sagittal plane of the skull. The medial face is defined by small bone projections that give it an irregular profile. The ventral face of the pars cochlearis is strongly convex in the transverse plane and longer than wide (W/L = 0.58). The promontorial groove is a shallow and anteromedially developed sulcus in the ventromedial surface of the pars coclearis. A small and oval foramen of uncertain homology opens in the ventral surface of the periotic, anterior to the mallear fossa. The compound posterior process of the tympanoperiotic is posterolaterally oriented forming an obtuse angle (~115°) with the anteroposterior axis of the pars cochlearis. Ventromedially, it presents a well-developed posterior pedicle for the articulation of the tympanic bulla. The dorsal face of the process is tightly articulated with the squamosal along its entire length; contacts the squamosal anteriorly through the posterior meatal crest forming a groove; contacts the exoccipital posteriorly through a visible suture (state SR4); and contacts the squamosal laterally through an interdigitated suture. The medial margin of the posterior process is concave. The stylomastoid fossa is deep and defined by ventral and dorsal margins; it extends from the posterior face of the pars cochlearis onto the dorsomedial surface of the base of the posterior process.

In posterior view (Figs. 11A and 11B), the fenestra rotunda is higher than wide (Table 3), and reniform in outline. In the medial margin of the fenestra rotunda are medial and lateral sulci separated by a rounded crest. Within the aperture of the fenestra rotunda, the basal part of the laminar gap for the basilar membrane is visible (*Geisler & Luo, 1996*). The caudal tympanic process is the posteriormost point on the pars cochlearis, lateral to the fenestra rotunda and medial to the facial sulcus. It is thin, triangular in posteromedial view and the dorsal surface is concave and posteroventrally oriented. The dorsal surface of the caudal tympanic process forms the floor of the stapedial muscle fossa,

A

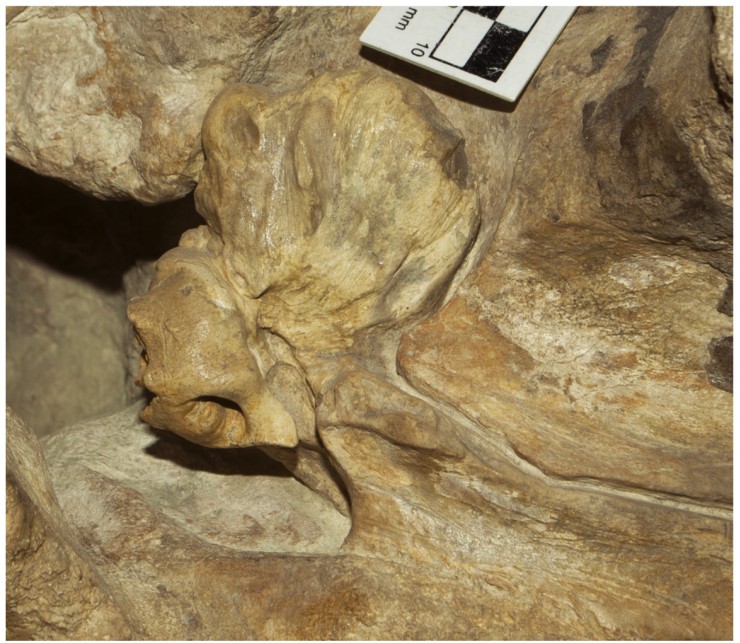

B

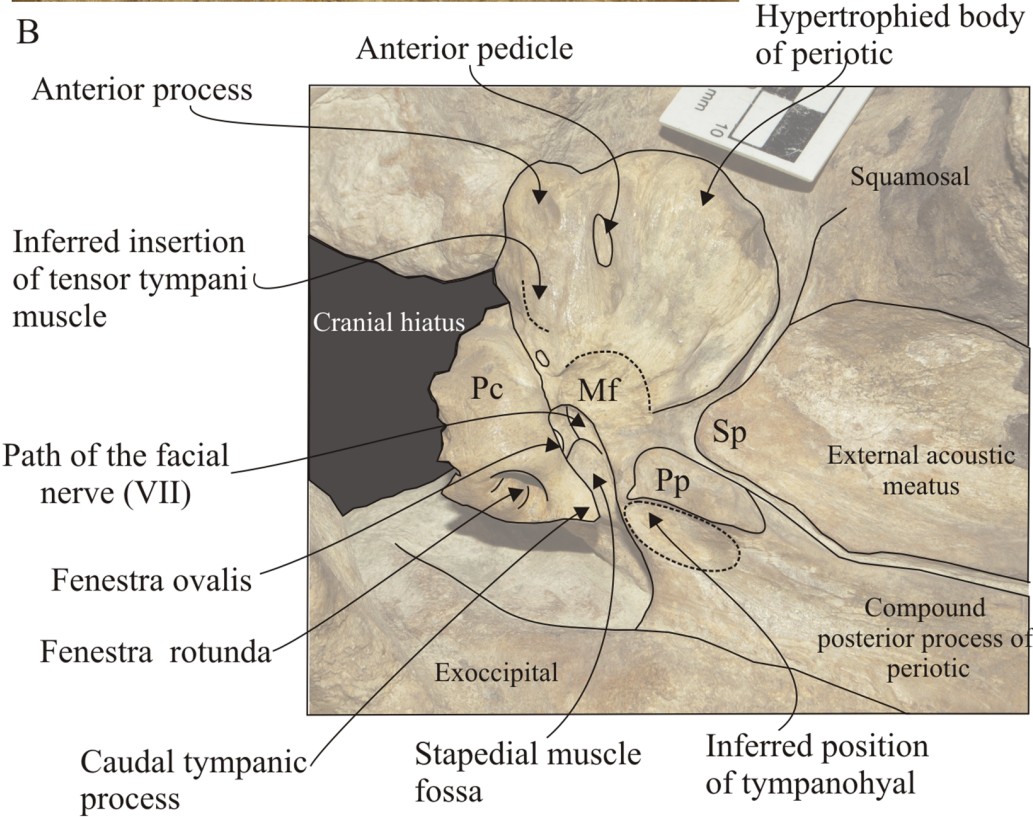

**Figure 10** *Morenocetus parvus*, **holotype, MLP 5–11, left periotic.** (A) Ventral view. (B) Key features in ventral view. Mf, mallear fossa; Pc, pars cochlearis; Pp, posterior pedicle; Sp, spiny process.

**Table 3 Measurements of the left periotic of the holotype of _Morenocetus parvus_, (MLP 5–11) in mm.**

| Measurement | MLP 5–11 |
|---|---|
| Length of the posterior process of the periotic | 118 |
| Length of anterior process of periotic | 31 |
| Width of anterior process of periotic | 12 |
| Maximum transverse width of the body of periotic | 34 |
| Anteroposterior diameter of pars cochlearis | 36 |
| Transverse width of the pars cochlearis | 21 |
| Dorsoventral diameter of the fenestra rotunda | 7 |
| Transverse diameter of the fenestra rotunda | 4 |

which is deep and oval-shaped. In lateral view part of the stapes can be seen in situ, occluding the aperture of the fenestra ovalis. The facial sulcus for the ventral path of the facial nerve (VII) continues as a deep groove onto the ventromedial side of the posterior process of the periotic. In dorsomedial view (Figs. 11C and 11D), the endocranial foramina are defined by the elevated dorsomedial rim of the promontorium. Anteriorly is the internal auditory meatus that is divided by a low transverse crest into two foramina: the smaller and circular proximal opening of facial nerve canal (VII); and the larger and circular foramen for the vestibulocochlear cranial nerve (VIII) (dorsal vestibular area). Both foramina are aligned anteroposteriorly. Posterior to the internal auditory meatus, is the small and rounded aperture for the cochlear aqueduct. The aperture for the cochlear aqueduct and the aperture for the endolymphatic duct are not aligned anteroposteriorly. Dorsolateral to the internal acoustic meatus is a well defined but shallow suprameatal fossa. This fossa is lined laterally by a well-developed crest that corresponds to the superior process.

## Comparative skull morphology

One of the most distinctive features of the skull of _Morenocetus_ is the orbital region that clearly differs from other balaenids. _Morenocetus_ has a transversely short and slightly ventrolaterally oriented supraorbital process of the frontal, whereas in _Balaenula_, _Balaenella_, _Balaena_ and _Eubalaena_ the supraorbital process is transversely long and strongly directed posteroventrally. The orbit of _Morenocetus_ is positioned relatively high on the cranium (viewed laterally, half the vertical distance between the vertex of the skull and the base of the postglenoid process), unlike other balaenids where the orbit is positioned lower on the cranium (see Discussion). The optic canal is developed only in the posterior two-thirds of the supraorbital process, unlike the condition observed in the other balaenids where the optic canal occupies the entire anteroposterior width of the ventral surface of the process. In addition, the canal remains open along its entire length, whereas in _Balaena mysticetus_, _Balaenula_ sp. and _Eubalaena_ spp. the optic canal is ventromedially enclosed. Another striking character is the lack of a narial process of the frontal separating the posterior portions of the nasals as observed in all remaining balaenids.

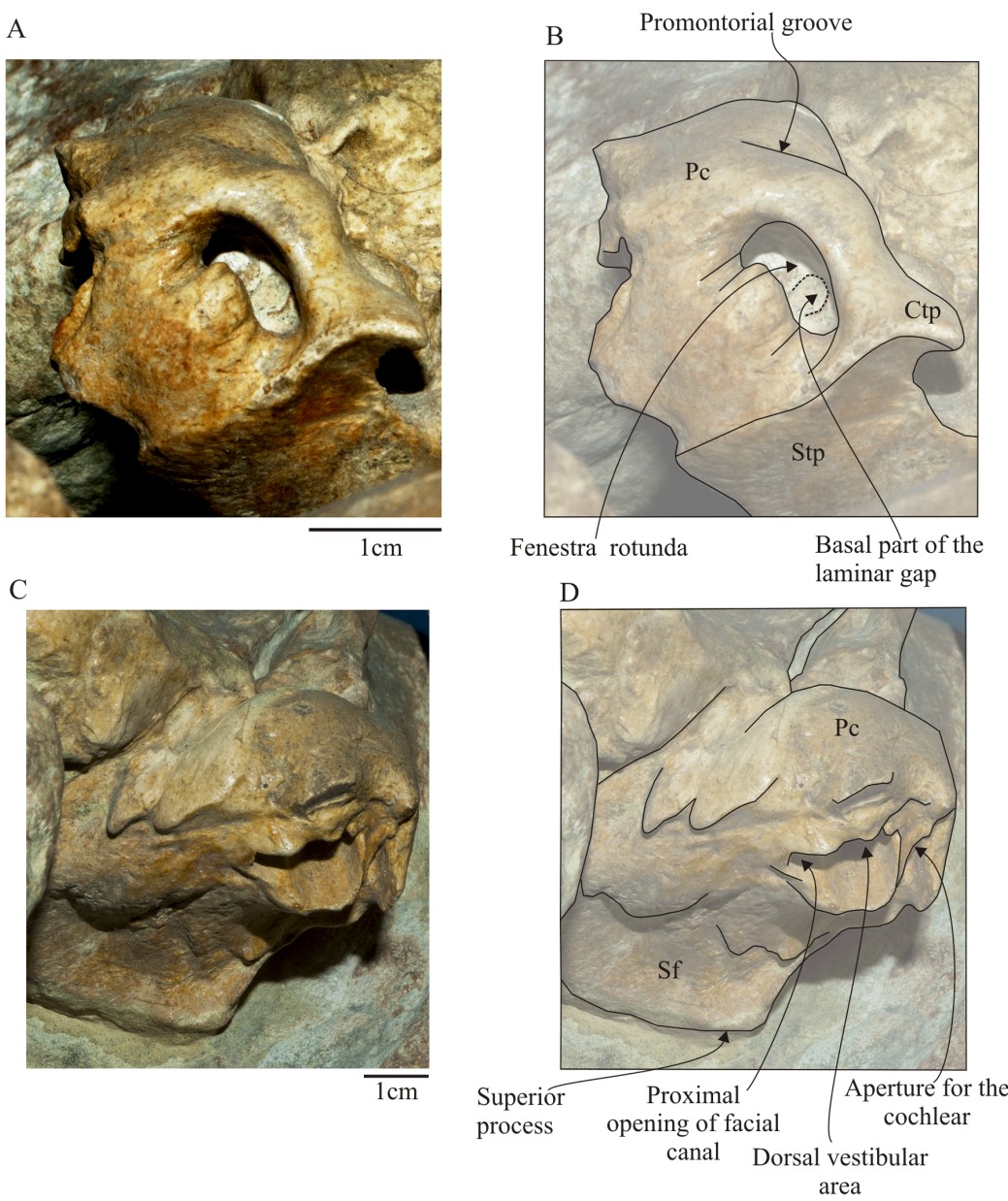

**Figure 11 *Morenocetus parvus*, holotype, MLP 5–11, left periotic.** (A) Posterior view. (B) Key features in posterior view. (C) Dorsomedial view. (D) Key features in dorsomedial view. Pc, pars cochlearis; Ctp, caudal tympanic process; Sf, suprameatal fossa; Stf, stylomastoid fossa.

The temporal region of *Morenocetus* also differs in morphology from other balaenids. In *Morenocetus* the parietal forms an extensive area of the external surface of the braincase, covering the posteromedial corner of the supraorbital process of the frontal and, in lateral view, the frontal–parietal suture is concave posteriorly as in *Balaenula astensis*. In contrast, in *Balaena* and *Balaenella* the spreading of the parietal is less pronounced and the suture between the frontal and parietal is almost straight. In *Eubalaena australis* the spreading of parietal is very pronounced with a well-defined anteromedial projection,

and the fronto-parietal suture is sinusoidal. The squamosal prominence is present but less developed than in *Eubalaena* spp. and *Balaena mysticetus*. The apex of the zygomatic process is ventrally deflected as in *Balaenula* sp., but unlike the remaining balaenids. The postglenoid process is ventrally oriented in posterior view whereas in *Balaena mysticetus*, *Balaenula astensis*, and *Eubalaena* it is ventromedially directed.

The occipital region of *Morenocetus* differs from other balaenids by having: an occipital shield that is long and triangular-shaped with straight lateral margins (unlike other balaenids where the lateral margins are convex); and a short exposure of the squamosal lateral to the exoccipital (all other balaenids have a broadly exposed squamosal lateral to the exoccipital).

In the basicranium of *Morenocetus*, the pterygoid sinus fossa is located anterior to the foramen pseudovale unlike the condition in all other balaenids where the pterygoid sinus fossa extends to a point approximately in line with the anterior edge of the foramen pseudovale. The periotic of *Morenocetus* has a short anterior process (similar to *Peripolocetus*) whereas in *Balaenula* sp., *Balaena mysticetus* and *Balaena ricei*, as well as extant species of *Eubalaena*, the anterior process is longer than the pars cochlearis. *Morenocetus* lacks a lateral tuberosity of the periotic (similar to *Peripolocetus*) whereas in remaining balaenids the lateral tuberosity is distinctly hypertrophied. The foramina of the internal acoustic meatus and aperture for the cochlear aqueduct are anteroposteriorly aligned unlike *Balaena mysticetus* and *Balaenula* sp. The suprameatal fossa is not hypertrophied in *Morenocetus*, a feature shared with *Peripolocetus*, but unlike *Balaena* and *Eubalaena*. The compound posterior process is oriented posterolaterally (forming an obtuse angle; ~115°) with respect to the long axis of the anterior process (with the periotic in situ), a condition shared with *Balaena* and *Peripolocetus*, but unlike *Eubalaena*, *Balaenella* and *Balaenula* where it forms a right angle with the anterior process.

## Phylogenetic analysis
### Morphological data
The equal weight parsimony analysis of the morphological data partition recovered 3 MPTs of 842 steps [consistency index = 0.41; retention index = 0.72] (Fig. 12A). Application of IterPCR procedure over the entire set of MPTs did not identify any taxa as unstable.

The results of our morphological phylogenetic analysis are congruent with the results of previous morphological and total evidence studies in recovering Balaenidae as the sister group of Plicogulae (*Fordyce & Marx, 2013*; *Gol'din & Steeman, 2015*; *Marx & Fordyce, 2015*). Within chaeomysticetes, Balaenidae is recovered as a well supported group based on the following synapomorphies: posterior border of zygomatic process of the squamosal and lateral edge of exoccipital confluent (67:1); zygomatic process of squamosal anterolaterally directed (86:2); squamosal higher than long (92:1); body of periotic lateral to pars cochlearis hypertrophied lateroventrally (146:1); lateral furrow on the tympanic bulla oriented anteroventrally (178:1); lack of a crest connecting medial and lateral lobes of the tympanic bulla (188:1); anteriormost point of the involucral ridge in line with the anterior border of the bulla (189:1).

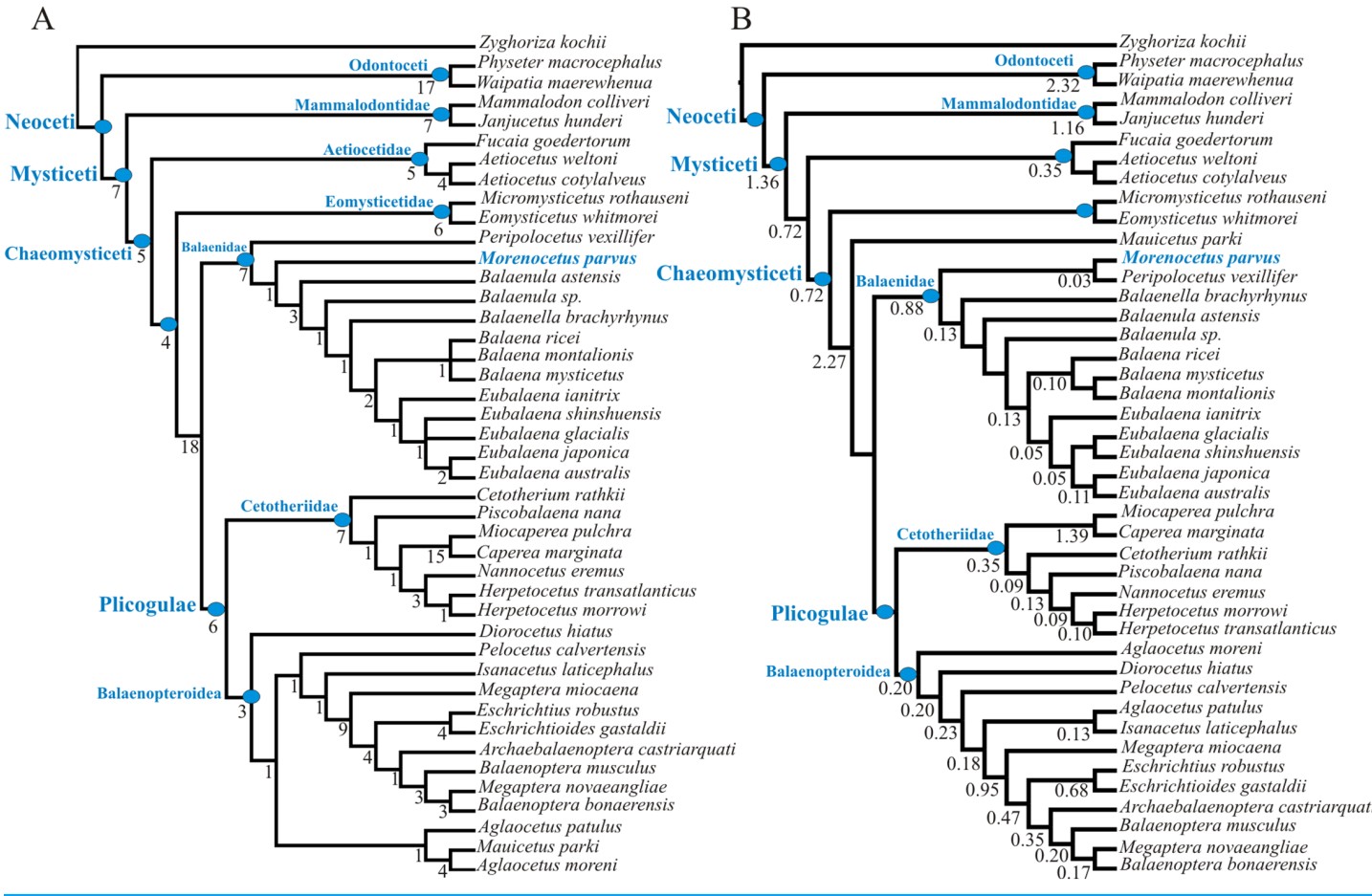

**Figure 12 Phylogenetic relationships of *Morenocetus parvus* based on morphological analysis.** (A) Strict consensus tree of equally weighted analysis (842 steps; CI = 0.41; RI = 0.72). (B) Single most parsimonious tree of implied weighting analysis (*K* = 3; fit = 83.13). Numbers at nodes are decay indices.

In contrast to the phylogenetic analysis of *Marx & Fordyce (2015)*, and in agreement with *Gol'din & Steeman (2015)*, *Peripolocetus* is recovered as the earliest diverging balaenid. Contrary to previous interpretations of *Morenocetus* as a taxon deeply nested within Balaenidae (*Bisconti, 2000, 2005*), forming a polytomy at the base of Balaenidae (*Churchill, Berta & Deméré, 2012*), or even outside Balaenidae (*Bisconti, Lambert & Bosselaers, 2017*), *Morenocetus* was estimated as sister to all balaenids more crownward than *Peripolocetus* in all MPTs. *Morenocetus* possesses several features that unequivocally identify it as a balaenid: the posterior margin of the zygomatic process of squamosal and lateral edge of the exoccipital forming a continuous lateral skull border (character 67:1); anterolaterally directed zygomatic process of the squamosal (character 86:2); squamosal that is higher dorsoventrally than long anteroposteriorly (character 92:1); and the body of the periotic lateral to the pars cochlearis is laterally and ventrally hypertrophied (character 146:1). The recent hypothesis that *Morenocetus* falls outside Balaenidae *sensu stricto* (*Bisconti, Lambert & Bosselaers, 2017*) is not supported by our results. The latter authors' phylogenetic hypothesis rests on 17 putative synapomorphies (only three unambiguous),

none of which can be scored in *Morenocetus* because they are characters of bones currently unknown for this taxon (tympanic bulla, rostrum, mandible and vertebrae).

*Morenocetus* and *Peripolocetus* share a combination of characters in the periotic that, according to this analysis, are interpreted as plesiomorphic: anterior process of the periotic shorter than the anteroposterior length of the pars cochlearis (character 139:0); lack of a lateral tuberosity of the periotic (character 144:0); suprameatal area of the periotic not hypertrophied (character 162:0); and compound posterior process of the tympanoperiotic oriented posterolaterally with respect to the longitudinal axis of the anterior process (character 170:0). However, *Morenocetus* is more specialized than *Peripolocetus* in having a zygomatic process of the squamosal dorsoventrally expanded (character 85:1); a dorsal extension of the tensor tympani on the medial side of the anterior process (character 149:1) and a narrow exposure of the squamosal lateral to the exoccipital (character 95:1). The incompleteness of the *Peripolocetus* holotype prevents a more detailed comparison with *Morenocetus*. Although IterPCR analysis did not identify *Peripolocetus* or *Morenocetus* as unstable taxa, it must be noted that the completeness of character scoring in these taxa is relatively low (77 and 55% missing data, respectively) and that the branch support for the node from which *Peripolocetus* and *Morenocetus* diverge is low. Thus, more information provided by new specimens of *Peripolocetus* (*Deméré & Pyenson, 2015*) as well as a new late Miocene balaenid from Argentina (M.R. Bueno, 2017, unpublished data) might change the proposed phylogenetic position of these taxa.

One synapomorphy supported the *Morenocetus* + *Balaenula* + *Balaenella* + *Balaena* + *Eubalaena* clade: zygomatic process of squamosal higher dorsoventrally than wide transversely (85:1). In *Peripolocetus* the zygomatic process is broken and the morphology of the base suggests that it is not dorsoventrally expanded. In contrast, the zygomatic process of *Morenocetus* was reassessed in this analysis based on our study of the referred specimen (MLP 5–15). In this taxon, as well as in remaining balaenids, the zygomatic process is dorsoventrally expanded.

*Balaenula astensis* was recovered as the sister taxon of the clade including *Balaenula* sp., *Balaenella*, *Balaena* and *Eubalaena* based on the following synapomorphies: width of the supraorbital process of the frontal more than twice the length above the orbit (37:2); posterior portion of the nasals separated posteriorly along the cranial midline by narial process of frontal (66:0); lack of paired tubercles on the supraoccipital (99:0); compound posterior process of the periotic oriented at a right angle to the axis of the anterior process (170:1, with reversal to the state 0 in *Balaena mysticetus* and *Balaena montalionis*). This sequence of clade divergence differs from *Bisconti (2005)*, *Churchill, Berta & Deméré (2012)*, *Marx & Fordyce (2015)* and *Bisconti, Lambert & Bosselaers (2017)*. The paraphyly of *Balaenula* spp. reflects the need for a modern thorough anatomical description and taxonomic revision of specimens referred to this genus. The re-interpretation of many characters of the vertex of *Balaenella* (*Marx & Fordyce, 2015*) and the increased taxonomic sampling of balaenids (this analysis) results in a new phylogenetic position of *Balaenella* supported by: anterior edge of supraorbital process posteriorly oriented (character 31:1) and orbital rim thickened with a rounded lateral surface (character 40:2). Finally, a clade

including a polytomy of *Balaena* and a *Eubalaena* spp. clade is supported by the following synapomorphies: parieto-squamosal suture forms a crest (93:1); compound posterior process of the periotic not exposed on lateral skull wall (172:0); and well-defined transverse creases on the dorsal surface of the involucrum of the tympanic bulla (191:1). The *Eubalaena* clade is diagnosed by one synapomorphy: foramen pseudovale raised above the lateral portion of the squamosal (119:1). The relationships amongst *Eubalaena* species are resolved, with *Eubalaena belgica* placed as sister to a clade including *Eubalaena glacialis*, *Eubalaena shinshuensis*, *Eubalaena japonica* and *Eubalaena australis*. In agreement with *Churchill, Berta & Deméré (2012)*, *Eubalaena japonica* and *Eubalaena australis* were recovered as sister taxa based on one synapomorphy: body coloration with the presence of a dorsal blaze (256:1). This analysis failed to resolve relationships within *Balaena*, but its monophyly is supported by one synapomorphy: portion of rostrum anterior to nasals raised above the level of the supraoccipital (2:2).

The implied weight analysis ($K = 3$) results in four MPTs with a fit of 83.13. The strict consensus tree depicts *Morenocetus* forming a clade with *Peripolocetus* (Fig. 12B). Unlike equal weight parsimony analysis, the internal relationships of remaining balaenids were not more clearly established. The implied weight analyses with $K = 6$ and $K = 10$ (six MPTs; fit of 54.73 and 37.82, respectively) result in the same topology obtained under equal weight analysis, except in the recovery of *Morenocetus* in a basal polytomy with *Peripolocetus* and a clade of all other balaenids.

### Combined data

The parsimony analysis of the combined matrix in TNT resulted in 24 trees with a tree length of 7,603. The total-evidence analysis does not result in a better resolution of relationships within Balaenidae (Fig. S2)

## DISCUSSION

### Specimens referred to *Morenocetus* by *Cabrera (1926)*

In the original description of *M. parvus*, *Cabrera (1926)* referred other materials to this species, two of which (i.e., MLP 5–30 four cervical vertebrae and MLP 5–21 a mandible) are not figured. As has been previously mentioned, the cervical vertebrae were not located in the La Plata Museum collection and the collagen and nitrogen analysis of the mandibles of MLP 5–21 indicate that this specimen is, at least, not Miocene, and therefore could not be confidently referred to *M. parvus*.

The specimen MLP 5–21 is clearly identified as a Balaenidae based on the twisted anterior end of the mandible and the presence of a mylohyoidal sulcus. One of the most conspicuous characters of this specimen is the presence of a very broad coronoid process, which is very similar to neonate and juvenile specimens of *Eubalaena australis*. The morphology of the coronoid region of MLP 5–21 has been discussed in *Marx et al. (2013)* in order to shed light on the debated relationship between balaenids and *Caperea*. In this sense, the coronoid morphology observed in the balaenid indet. MLP 5–21 and juvenile specimens of *Eubalaena australis* are very different to those observed in *Caperea* juveniles (*Marx et al., 2013*). Although MLP 5–21 cannot be referred to *Morenocetus*, it is

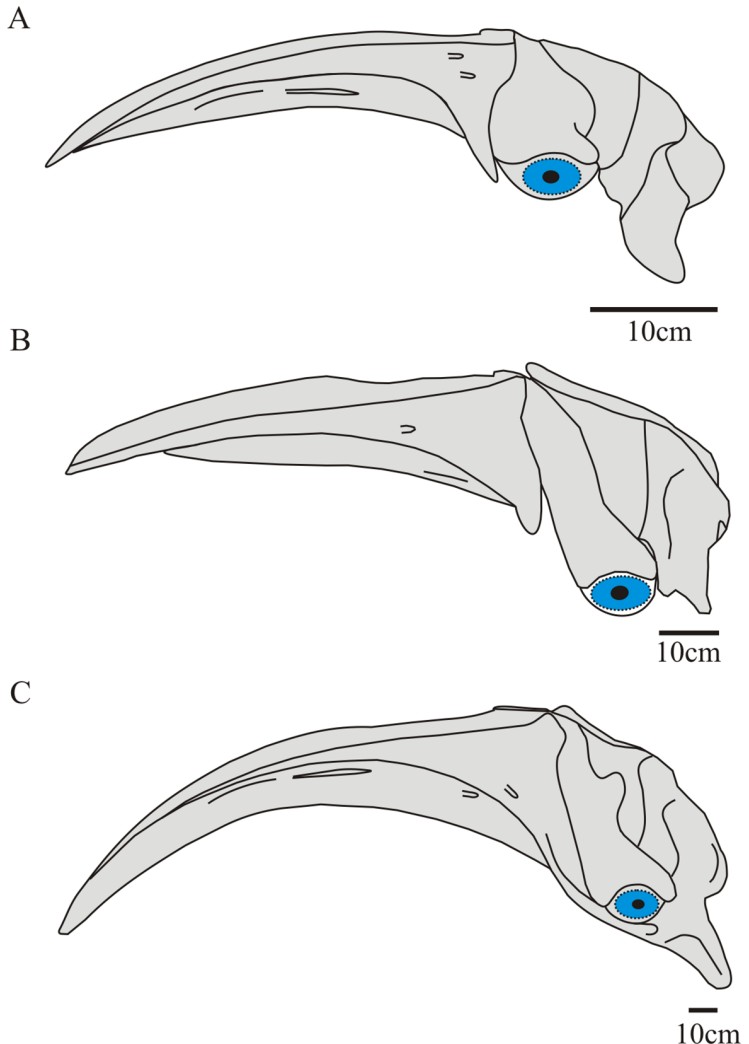

A

10cm

B

10cm

C

10cm

**Figure 13 Reconstructions of balaenid crania in left lateral view, showing relative position of the orbit and eyeball (in blue) with arching of the rostrum.** (A) *Morenocetus parvus.* (B) *Balaenella brachyrhynus.* (C) *Eubalaena australis.* The rostrum in (B) is depicted as preserved in the original specimen.

clearly a balaenid with a coronoid morphology that could be considered close to *Eubalaena* juveniles, and completely different to *Caperea*, giving additional support to the hypothesis that neobalaenines are not closely related to balaenids.

## Orientation of the skull

The orientation of the skull of extinct balaenids has been problematic due to the arching of the rostrum. To this it must be added that, in most of the fossil balaenid specimens, the rostrum is not preserved. Incorrect skull orientation in the past may have misled some character interpretations and, consequently, their scoring for phylogenetic analysis. Cranial reconstructions based upon application of the proposed landmarks are summarized in Fig. 13. The correct orientation of the skull affects the topography of bones and sutures in the temporal and orbital region, and also the reconstruction of rostrum

shape. The orientation of the squamosal depends strongly on the orientation of the cranium as a whole. Among balaenids, the squamosal is posteriorly directed in *Balaena* species whereas it is anteriorly directed in *Morenocetus*, *Balaenula*, and *Eubalaena*. In *Balaenella* the squamosal has been traditionally interpreted as posteriorly directed (and hence this character has been proposed to support a close relationship between *Balaena* and *Balaenella*) (*Bisconti, 2005*; *Churchill, Berta & Deméré, 2012*). A re-orientation of the skull of *Balaenella* according to the landmark proposed here allows a re-interpretation of this character (i.e., the squamosal is anteriorly directed; Fig. 13B).

Another interesting result of the orientation proposed herein relates to the orbit location. The orbits of *Morenocetus* were located approximately at half the vertical distance between the vertex of the skull and the base of the postglenoid process (Fig. 13A). This position is relatively high in comparison to other balaenids, and similar to that of Balaenopteridae and *Caperea*. Thus, in *Balaenella*, *Balaenula astensis*, *Balaenula balaenopsis*, extant *Eubalaena* and *Balaena mysticetus* the orbits are placed at 2/3 the vertical distance from the vertex of the skull to the base of the postglenoid process (Figs. 13B and 13C). In addition, the position of the postorbital process of the frontal is very close to the zygomatic process of the squamosal. As orbits have not been preserved in *Peripolocetus*, it is not clear if this condition is unique to *Morenocetus* among stem balaenids. Also in *Morenocetus*, the orbits are more laterally directed, which differs from the posteroventral direction of the orbits in Pliocene–Recent balaenids.

Noteworthy changes in the orbit orientation appear tightly coupled to an increase in the arching of the rostrum and the vertical orientation of the occipital shield. The reconstruction of the rostrum of *Morenocetus* suggests it is moderately arched dorsoventrally while in extant balaenids it is strongly arched dorsoventrally (Fig. 13). In the same way, changes of the vertical orientation of the occipital shield are evident in the skull when viewed laterally. In *Morenocetus* the occipital shield is oriented forming an acute angle with the anteroposterior axis of the skull (~25°) while in extant balaenids, the occipital shield is more vertically oriented, this angle being ~35°.

The evolution of skull structure in balaenids, characterized by a general trend toward the arching of the skull, has functional implications for vision. Experimental and behavioral observations on both odontocetes and mysticetes demonstrate that, contrary to traditional hypotheses (*Weber, 1886*; *Walls, 1942*), vision plays an important role for orientation and navigation, coordination of group movements, identification of conspecific individuals, and communication (*Madsen & Herman, 1980*). In Cetacea field of vision is permitted by smooth eyeball movements by means of completely developed oculomotor muscles and nerves (*Dawson, 1980*; *Madsen & Herman, 1980*) and fat surrounding the eyeballs (*Ninomiya & Yoshida, 2007*; *Buono, Fernández & Herrera, 2012*). The eyeball movements, coupled with a peculiar retinal organization with two best vision areas, compensate for the restricted mobility of the head of cetaceans and provide acute vision in various parts of the visual field (*Mass & Supin, 1995*, *1997*). Reconstruction of the *Morenocetus* skull, and comparison with the skull profile of more advanced balaenids, indicates that the increase in skull arching throughout Balaenidae evolution would have reduced the field of vision if the bony orbits were to remain in a high

position on the cranium, as exemplified by *Morenocetus*. A functional resolution to compensate for this trade-off could have been the ventral displacement of the orbits and eyeballs in crown Balaenidae.

### Morenocetus: a primordial balaenid?

With an early Miocene age, *Morenocetus* is the oldest balaenid so far reported with the potential to provide insights into the origin of specialized morphology of right whales. *Morenocetus* exhibits primitive characteristics that support its basal position within balaenids, such as a short and rounded anterior process of the periotic; lack of a lateral tuberosity and a hypertrophied suprameatal area, a compound posterior process of the tympanoperiotic oriented posterolaterally with respect to the longitudinal axis of the anterior process, an optic canal ventrally open; a pterygoid sinus fossa extending anterior to the foramen pseudovale. In addition, *Morenocetus* clearly differs from geologically younger and more crownward balaenids (i.e., *Balaenula*, *Balaenella*, *Balaena* and *Eubalaena*), in having a relatively narrow and short supraorbital process and in the lack of a narial process separating the posterior margins of the nasals, thus supporting its early divergence within balaenids.

Yet although *Morenocetus* possesses some primitive features of the cranium, it exhibits derived morphology uniquely shared with other balaenids and rather disparate from all other chaeomysticetes. The periotic morphology clearly differs from described stem Chaeomysticeti and closely resembles modern right whales in the presence of a hypertrophied body of the periotic, an indistinct mallear fossa, and a distinct ridge delimiting the insertion surface for the tensor tympani muscle. In addition *Morenocetus* shares with modern balaenids a dorsoventrally expanded zygomatic process of the squamosal, a parieto-squamosal suture shaped like a crest, a foramen pseudovale located within the squamosal and opening posteriorly, and a short ascending process of the maxilla. Given the geologic antiquity of *Morenocetus*, it is clear that specialized morphological traits that characterize all living balaenids were acquired early in the evolutionary history of the clade, without significant changes between the early Miocene and Recent.

The geologic age and phylogeny of *Morenocetus* has bearing on the broader evolution of the major mysticete clades. The origin of living families of baleen whales has been estimated around the late Oligocene based on molecular divergence data (*Steeman et al., 2009*; *Marx & Fordyce, 2015*; *McGowen, Spaulding & Gatesy, 2009*). However the fossil record of mysticetes reveals an important late Oligocene–early Miocene gap (Fig. 14). Mysticete fossils from this stratigraphic interval are globally scarce (Table 4). With an early Burdigalian age, *Morenocetus* is among the oldest crown Mysticeti known, reinforcing the idea that the timing for the diversification for crown lineages must have occurred no later than the late Oligocene. Strikingly, stem balaenopteroids are better represented than stem balaenids for the late Oligocene–early Miocene (Fig. 14). It is not clear if this gap in the balaenid fossil record is real or reflects uneven sampling. Molecular sequence data have been used to estimate that balaenids diverged during the late Oligocene (29–26 Ma, *Steeman et al., 2009*; 23 Ma, *Marx & Fordyce, 2015*), implying that fossils of stem balaenids should occur in strata dating to this time. The study of late

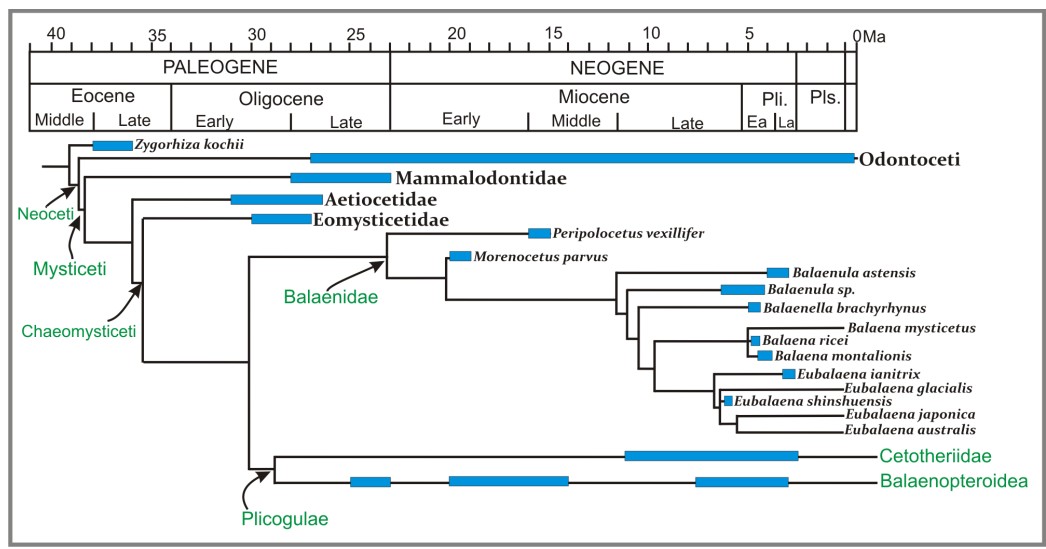

**Figure 14 Stratigraphically calibrated phylogeny of Balaenidae.** Time calibrated phylogeny of Balaenidae pruned from our consensus cladogram in Fig. 12A with Odontoceti, Mammalodontidae, Aetiocetidae, Eomysticetidae and Plicogulae collapsed. Stratigraphic range data derived from published accounts for each taxon (See Table S2, for source of age data), calibration for major nodes follows mean divergence date estimates by *Marx & Fordyce (2015)*: Table 1*)* (Neoceti = 38.80 Ma; Mysticeti = 38.42 Ma; Chaeomysticeti = 35.5 Ma; crown Mysticeti 30.35 Ma; Balaenidae = 23.08 Ma; crown Balaenidae = 9.82; Plicogulae = 28.96 Ma).

**Table 4 Early Miocene (Aquitanian-Burdigalian) fossil records of mysticetes.**

| Taxon | Age | Reference |
| --- | --- | --- |
| *cf. Waharoa* OU 22744 | Aquitanian (22.28–22.8 Ma); Hakataramea Quarry, New Zealand | *Boessenecker & Fordyce (2017)* |
| Unnamed balaenopteroid ZMT 67 | Aquitanian (23–21.7 Ma); South Canterbury, New Zealand | *Marx & Fordyce (2015)* |
| Unnamed balaenopteroid OU 22705 | Aquitanian (21.7–20.5 Ma), South Canterbury, New Zealand | *Marx & Fordyce (2015)* |
| *Aglaocetus moreni* (MLP 5–1; MLP 5–14; FMNH 13407) | Burdigalian, Cerro Castillo, Chubut, Argentina | *Lydekker (1894)*, *Cabrera (1926)*, *Kellogg (1934)*; this study (see Geological context for a discussion of the age of Gaiman Formation) |
| *Diorocetus chichibuensis* | Late Burdigalian–Langhian (16.4–15.1 Ma), Saitama Prefecture, Japan | *Marx & Fordyce (2015)* |
| *Diorocetus shobarensis* | Late Burdigalian–Langhian (17–4.9 Ma), Hiroshima Prefecture, Japan | *Marx & Fordyce (2015)* |
| *Isanacetus laticephalus* | Late Burdigalian (17.5–16 Ma), Gifu Prefecture, Japan | *Marx & Fordyce (2015)* |
| *Parietobalaena palmeri* | Late Burdigalian–Langhian (16.4–14.5 Ma), Maryland, Virginia, USA | *Marx & Fordyce (2015)* |
| *Parietobalaena yamaokai* | Late Burdigalian–Langhian (17.0–14.9 Ma), Hiroshima Prefecture, Japan | *Marx & Fordyce (2015)* |
| *Titanocetus sammarinensis* | Late Burdigalian–Langhian (16.4–14.7 Ma), Monte Titano, Republic of San Marino | *Marx & Fordyce (2015)* |
| Unnamed aetiocetid OCPC 1178 | Early–late Burdigalian (18.8–17.2 Ma), Orange County, USA | *Marx & Fordyce (2015)* |

Oligocene specimens, previously reported as putative balaenids (*Fordyce, 2002*), may help to fill the gap in the evolutionary history of the group and confirm whether all stem balaenids were characterized by the surprisingly modern-looking morphology described here in *Morenocetus*.

## Evolution of Balaenidae body size

The estimated TL of *Morenocetus* (<6 m) is smaller than that obtained in Pliocene balaenids (e.g., *Balaenella*, *Balaenula astensis*, *Balaena montalionis* and *Eubalaena shinshuensis*) (6–12 m). These estimated TLs approximate the body length estimates of *Bisconti, Lambert & Bosselaers (2017)* for *Morenocetus* (4.8–6.2 m), *Balaenula astensis* (4.8–6.4 m), *Balaena montalionis* (7.5–10 m), *Eubalaena shinshuensis* (9.6–12.8 m), *Balaenella* (3.3–4.4 m) and *Eubalaena ianitrix* (5–7 m). The most remarkable differences between our estimates of body length and those of *Bisconti, Lambert & Bosselaers (2017)* are in *Balaenella* (3.3–4.4 m) and *Eubalaena ianitrix* (5.0–7.0 m). The previous estimate of body length in *Balaenella* is probably erroneous because it is based on skull length (*Bisconti, Lambert & Bosselaers, 2017*), which may yield underestimated values due to uncertain orientation of the cranium and incomplete preservation of the rostrum. In the case of *Eubalaena ianitrix*, the discrepancy in body length estimates could be related to the different approaches used in each analysis. The estimated body length for *Eubalaena ianitrix* by *Bisconti, Lambert & Bosselaers (2017)*, has a value very close to the body length estimated in this study for *Morenocetus*. However, comparisons of the BZW of both taxa, shows an important difference in skull size (*Morenocetus* BZW = 570 mm; *Eubalaena ianitrix* BZW = 1,660 mm). It is unlikely that both taxa with such disparity in skull size occupied a similar range of body length. Currently, there are no published equations that are based on a mysticete cranial measurement dataset including a large sample of balaenid specimens. It is outside the aims of this study to assemble such a dataset. We therefore used two different published equations by *Pyenson & Sponberg (2011)* and *Lambert et al. (2010)* to estimate body size. As discussed by *Bisconti, Lambert & Bosselaers (2017)* there are some limitations to these approaches due to inadequate sampling of TL + cranial metrics data from Balaenidae. Nonetheless, because our results using two different approaches are very close (one of them based on regression equations including extant balaenids; *Lambert et al., 2010*), and also our estimations yield similar results (with some exceptions) and generally approximate the estimates obtained by *Bisconti, Lambert & Bosselaers (2017)* using different proxies (i.e., supraoccipital length; skull/body ratio), we think that our body size estimates from these two approaches are plausible.

Traditionally, an increase in body size has been cited as a major unidirectional evolutionary trend within Mysticeti (*McLeod, Whitmore & Barnes, 1993*; *Fordyce & Barnes, 1994*; *Fordyce & de Muizon, 2001*; *Sanders & Barnes, 2002*; *Pyenson & Sponberg, 2011*; *Pyenson & Vermeij, 2016*; *Tsai & Kohno, 2016*; *Slater, Goldbogen & Pyenson, 2017*). However, optimization on recently estimated phylogenies suggest that considering a gradual increase in body length, from small-toothed mysticetes to large body size in Chaeomysticeti, is an oversimplification (*Fitzgerald, 2010*; *Tsai & Kohno, 2016*). Among balaenids, the oldest record of gigantism is represented by the late Miocene–early Pliocene

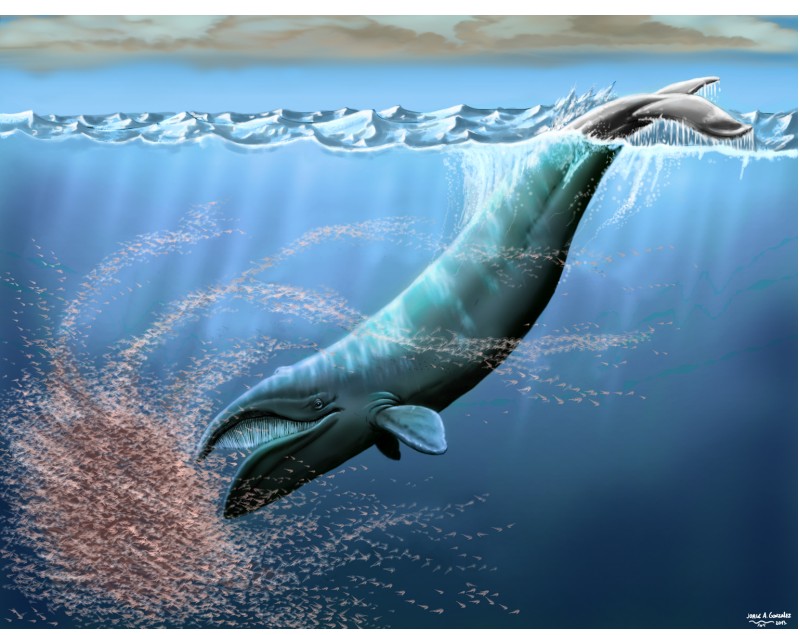

**Figure 15 Artistic restoration of *Morenocetus parvus* in life.** Art by Jorge Gonzalez.

balaenid *Eubalaena shinshuensis*. In the particular case of balaenids, a broad outline of body size evolution suggests that the primitive condition was a relatively small body length represented by *Morenocetus* (Fig. 15). Although the body length for the middle Miocene *Peripolocetus* cannot be estimated, comparable preserved elements of the latter taxon and *Morenocetus* suggests *Peripolocetus* had a body length similar to that of *Morenocetus*. Throughout the Pliocene, fossil records document the co-occurrence of balaenid taxa of different sizes ranging approximately from 6 to 12 m with the last occurrence of small forms (i.e., *Balaena montalionis*) by the end of the Pliocene. This pattern is consistent with a recent analysis of body size evolution in mysticetes, which suggested that the attainment of gigantic body sizes is a recent event (Plio-Pleistocene) in mysticete history (*Slater, Goldbogen & Pyenson, 2017*).

## Pending issues

Improvements to knowledge of balaenid evolution may be driven both by new discoveries in the field as well as by reexamination of fossils already in museum collections. In this sense, one of the most striking features of the fossil record of balaenids is the apparent scarcity of Miocene fossils. This contrasts with the large number of balaenopteroid mysticete taxa described from the Miocene, especially in the northern hemisphere. Is the sparse record of balaenids in the Miocene a real biological signal or is it merely an artifact of uneven collection effort? During the last century, richly fossiliferous marine Miocene strata of Patagonia, New Zealand and Australia have been scarcely prospected (*Fitzgerald, 2004*, *2012*; *Buono et al., 2014*; *Boessenecker & Fordyce, 2017*). On the contrary, Northern Hemisphere Miocene strata have been extensively explored for fossil cetaceans since the first decades of the 20th Century (*Barnes, 1976*; *Kellogg, 1965*, *1968*, *1969*).

A possible scenario is that the scarce fossil record of balaenids reflects a real evolutionary pattern and that the comparatively low diversity of the group during the Miocene mirrors their extant low diversity. The apparent increase of balaenid diversity during the early Pliocene (*Kimura, 2009*) should be cautiously interpreted as a biological signal, due to some historically significant European forms, including *Balaenula balaenopsis* and *Balaena primigenia*, needing modern taxonomic revisions.

## INSTITUTIONAL ABBREVIATIONS

**CNPMAMM**  Laboratorio de Mamíferos Marinos, Centro Nacional Patagónico, Puerto Madryn, Chubut, Argentina.

**FMNH**  The Field Museum of Natural History, Chicago, USA.

**MLP**  Museo de La Plata, La Plata, Argentina.

**NMNZ**  Museum of New Zealand Te Papa Tongarewa, Wellington, New Zealand.

**OCPC**  The Cooper Center, Orange County, USA.

**OU**  Geology Museum, University of Otago, Dunedin, New Zealand.

**USNM**  National Museum of Natural History, Smithsonian Institution, Washington DC, USA.

**ZMT**  Fossil mammals catalogue, Canterbury Museum, Christchurch, New Zealand.

## ACKNOWLEDGEMENTS

We thank E. Crespo and N. Garcia (CESIMAR, Centro Nacional Patagónico), and C. W. Potter, D. J. Bohaska and N. D. Pyenson (National Museum of Natural History), R. E. Fordyce (Otago Museum), Marcelo Reguero and Alejo Scarano (Museo de La Plata) for allowing M. R. Buono and E. M. G. Fitzgerald to examine specimens in their care and for assistance during their visits. Juan Jose Moly and Leonel Acosta Burllaile (Museo de La Plata) are thanked for the preparation of the specimens. We thank F. Marx, R. E. Fordyce, M. Churchill and O. Lambert for providing photographs of balaenid specimens. We thank F. Marx and R. E. Fordyce for helpful discussions of *Morenocetus* and Balaenidae morphology. Dr. Eduardo Tonni is thanked for their help with collagen analysis of MLP 5–30. We are extremely grateful to the personnel and volunteers of the Stranding Network at Península Valdés (V. Rowntree, M. Sironi, M. Uhart, A. Chirife, M. Di Martino), Lucia Alzugaray, and H. Ruiz for their valuable help and support during dissections in the field and in the lab. Lucia Alzugaray is thanked for providing photographs of Eubalaena australis dissections. M. T. Dozo (CENPAT-CONICET) is thanked for her assistance during this study. J. Sterli (Mef) is thanked for her assistance with the combined phylogenetic analysis. The authors are grateful to Jorge Velez-Juarbe, Robert Boessenecker and an anonymous reviewer and editor (J. Thewissen); their suggestions greatly improved the quality of this article. This contribution used TNT version 1.5, a program made freely available thanks to a subsidy by the Willi Hennig Society. This article is part of doctoral research of Monica R. Buono at Universidad de La Plata and Centro Nacional Patagónico (CCT CONICET-CENPAT).

### Funding

This research has been supported by the following grants: Cetacean Society International (CSI), American Museum of Natural History (Lerner Gray Fund for Marine Research), Society for Marine Mammalogy (Grant In Aid of Research), Smithsonian Institution (Remington Kellogg Fund) and Agencia Nacional de promoción Científica y Tecnológica (grant number PICT 0682) to Mónica R. Buono.; Agencia Nacional de promoción Científica y Tecnológica (grant number PICT 0792) to Mónica R. Buono and José I. Cuitiño, and Agencia Nacional de promoción Científica y Tecnologica (grant number PICT 0748) to Marta S. Fernández; and an Australian Research Council Linkage Project (grant number LP150100403) to Erich M.G. Fitzgerald. The funders had no role in study design, data collection and analysis, decision to publish, or preparation of the manuscript.

### Grant Disclosures

The following grant information was disclosed by the authors:
Cetacean Society International (CSI).
American Museum of Natural History (Lerner Gray Fund for Marine Research).
Society for Marine Mammalogy (Grant In Aid of Research).
Smithsonian Institution (Remington Kellogg Fund).
Agencia Nacional de promoción Científica y Tecnológica: PICT 0682.
Agencia Nacional de promoción Científica y Tecnológica: PICT 0792.
Agencia Nacional de promoción Científica y Tecnologica: PICT 0748.
Australian Research Council Linkage Project: LP150100403.

### Competing Interests

The authors declare that they have no competing interests.

### Author Contributions

- Mónica R. Buono conceived and designed the experiments, performed the experiments, analyzed the data, contributed reagents/materials/analysis tools, wrote the paper, prepared figures and/or tables, reviewed drafts of the paper.
- Marta S. Fernández conceived and designed the experiments, performed the experiments, analyzed the data, contributed reagents/materials/analysis tools, wrote the paper, reviewed drafts of the paper.
- Mario A. Cozzuol conceived and designed the experiments, performed the experiments, analyzed the data, contributed reagents/materials/analysis tools, wrote the paper, reviewed drafts of the paper.
- José I. Cuitiño conceived and designed the experiments, analyzed the data, contributed reagents/materials/analysis tools, wrote the paper, prepared figures and/or tables, reviewed drafts of the paper.
- Erich M.G. Fitzgerald conceived and designed the experiments, performed the experiments, analyzed the data, contributed reagents/materials/analysis tools, wrote the paper, reviewed drafts of the paper.

## Field Study Permissions

The following information was supplied relating to field study approvals (i.e., approving body and any reference numbers):

Field dissections were approved by the Dirección de Fauna y Flora Silvestre del Chubut.

## Data Availability

The raw data (morphological matrix) is included as Supplemental Dataset Files.

## Supplemental Information

Supplemental information for this article can be found online at http://dx.doi.org/10.7717/peerj.4148#supplemental-information.

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
