# Peer review of "The early Miocene balaenid Morenocetus parvus from Patagonia (Argentina) and the evolution of right whales"

_PeerJ, doi:10.7717/peerj.4148_

## Round 0.1 · original submission · Minor Revisions

Two reviewers recommended minor changes, and one recommended major changes. All three reviewers provided a detailed list of issues that you need to consider and address. Some of these changes are mere wording issues, but others are serious and will require your attention. The reviewer who suggested that major revisions were in order listed four problematic areas:

1) Insufficient description of the characters used for the orientation of the skull;
2) Many character states used in the phylogenetic analysis are ordered but no explanation is provided suggesting that the authors had preconceived hypotheses of character evolution;
3) The equation used for body size reconstruction of Balaenidae was not developed for such a group thus results are expected to be wrong.
4) The English is not adequate and needs a thorough revision as it shows too many syntax errors and misspellings that, sometimes, make understanding difficult.

Of these, I consider 1 and 4 to be important issues to be addressed seriously. Regarding 2 and 3, you are explicit about what you have done, and I encourage you to improve the justification of your choices in the text, however, I do not believe that changes in these sections are required.

Reviewer 1 ·

Basic reporting

Authors' English is not adequate. The text is rich in syntax errors and misspellings and needs a profound revision by a native speaker. Use of commas and abbreviations is not consistent, names of authors cited in the literature are often misspelled. Editorial and linguistic revisions are highly needed.
No comment.
No comment.
The article is self-contained but some of the results cannot be accepted unless significant revisions of the methods. Suggested revisions are provided in the comments to the authors.

Experimental design

No comment.
No comment.
Problematic use of published methods to infer total body length in Balaenidae. Characters used for orientation of the skull insufficiently described. Suggested improvements are providedin the comments to the authors.
Method used for orientation of skull insufficiently described and impossible to replicate (acute angle not defined; posterior wall of nasal fossa not defined). Suggested improvements are provided in the comments to the authors.

Validity of the findings

Good validity as far as the anatomical description of Morenocetus parvus is concerned. Phylogenetic analysis must be supplemented with information about the authors' hypotheses of character evolution used to order a high number of character states. Body size reconstruction inadequate based on problematic use of the cited methodology. Suggested improvements are provided in the comments to the authors.
No comment.
Conclusion are well stated based on the methods used but such methods must be discussed and revised as far as body size reconstruction and phylogenetic analysis are concerned. Improvements are related to changes in the methods as suggested in the comments to the authors.
No comment.

Additional comments

This manuscript includes the re-description of an important fossil balaenid, that is Morenocetus parvus. The anatomical description is generally good, as only minor changes are needed to better explain the concepts presented by the authors. The organization of the manuscript is substantially good and the text does not need to be shortened, I only suggest moving the Comparative Skull Analysis before the Phylogenetic Analysis as it provides evidence then used to support the nodes found in the cladograms. The fossil described is important as it represents the oldest record of Balaenidae whose original description is from 1926 and is inadequate for being used in modern phylogenetic works.
Unfortunately, the manuscript shows several problems and, in my opinion, cannot be accepted for publication in its present form. I list such problems below and then discuss all of them:
1) Insufficient description of the characters used for the orientation of the skull;
2) Many character states used in the phylogenetic analysis are ordered but no explanation is provided suggesting that the authors had preconceived hypotheses of character evolution;
3) The equation used for body size reconstruction of Balaenidae was not developed for such a group thus results are expected to be wrong.
4) The English is not adequate and needs a thorough revision as it shows too many syntax errors and misspellings that, sometimes, make understanding difficult.
The English needs a thorough revision by a native speaker. Please, check verbs, their accordance with subjects; check commas in references; be consistent in your way to cite literature; check authors’ names (e.g., Kellogg includes two g). Please, do check also in the supplementary information files. In its current form, the English is highly incorrect. For example, on p. 8 lines 46 and following:
Balaenids has been consider a key group in understanding of baleen whales evolution, because is the oldest surviving lineage of crown Mysticeti, as suggested by their early…

This sentence must be changed into the following (or an equivalent):

Balaenids have been considered a key group in understanding baleen whale evolution because they represent the oldest surviving lineage of crown Mysticeti as suggested by their early

Materials
It is not clear why the referred specimen is not assigned to Morenocetus: is your decision based only on age assessment? Your decision must be based on morphology as you cannot be sure about species longevity for Morenocetus and you do not provide a quantitative assessment of the different age of the referred specimen. If it is not from Miocene, what is its age? How can you be sure about the fact that the observed differences in Nitrogen and collagen content exclude a Miocene age? Miocene spans from 23 to 5.3 million years ago: are the differences you found indicative of a difference in age of more than 15 million years?

Methods
Physical maturity: the study by Walsh & Berta (2011) was based on balaenopterids and it is not clear whether the chronology of suture closure that they detected on balaenopterids may be directly applied to Balaenidae. They applied to Balaenidae but, without a study of balaenid embryos an fetuses, their application to Balaenidae could be wrong. I strongly recommend authors to cite this problem in the direct application of Walsh & Berta (2011) method to Balaenidae.
Orientation of skull: Character 1 is not clear to me. What does it means that the posterior portion of the nasal fossa is perpendicular to the transverse axis of the skull? In lateral view? In dorsal view? Do you mean the anterior end of the nasal bones in dorsal view? Or do you mean the fundus of the nasal fossa (formed by ethmoturbinates) according to Mead & Fordyce (2009)? Character 2 depends on the whole orientation of the skull and may be a character typical of Eubalaena australis. How can use such a character for determining the orientation of different balaenid species? Character 3 must be defined more clearly. You must provide a value in degrees of the acute angle otherwise your character does not work. Acute angle is an angle included between 1° and 89°: please specify. Is Buono’s personal observation based on dissection? In this case, it is fundamental to understand your methodology otherwise character 3 cannot be accepted.
Phylogenetic analysis: The use of ordered characters suggests that the authors have an idea of how such characters evolved. In many papers, authors do not explain why they decided to treat some characters as ordered. I think that this should be avoided because it seems to me a way to get optimized phylogenetic searches based on preconceived hypotheses of character evolution. In my opinion, authors must explain why they treated characters 1, 4, 5, 25, 35, 37, 41, 58, 79, 86, 99, 104, 107, 114, 121, 172, 176, 182, 186, 204, 212, 226, 229, 231, 252 as ordered. This is a high number of ordered characters. Moreover, the authors use terms not described by their reference (Mead & Fordyce 2009): as an example, medial lobe of the tympanic bulla is not described there. Please, refer to the literature you cited or cite the relevant literature where the terms were defined.
Body size reconstruction: Pyenson & Sponberg (2011) developed the equation cited by the authors for stem Mysticeti that, in their work, included baleen whale species not belonging to crown Mysticeti. This means that the formula cannot be used to predict body size of Morenocetus, Balenella brachyrhynus, Balaena montalionis, Balaenula astensis and Eubalaena shinshuensis. The authors must remove all the parts of the manuscript related to this research and use different methods as the results are not based on the appropriate methodology.

Geological setting
It would be better to define authors’ concept of early Miocene. I guess that they mean Aquitanian-to-Burdigalian but more detail would greatly help in understanding. The authors cite U-Pb and Sr-Sr absolute datings: please cite the dates in million years to clarify the results of these analyses. A clear specification of the age of morenocetus parvus would be greatly helpful. I strongly recommend authors to change the generic ‘early Miocene’ date into a more detailed age horizon (e.g., Aquitanian, Langhian).

Emended diagnosis
If the total length was estimated based on the equation cited in the Methods section, it must be removed as the equation is not appropriate for the reconstruction (see above). Is ‘the narrow exposition of the squamosal’ observed in dorsal or lateral view? I suggest changing exposition into exposure. Bisconti (2000) published a redescription of Balaenula astensis showing what he called ‘lateral squamosal crest’ that corresponds to the supramastoid crest of Mead & Fordyce (2009). The authors stated that the supramastoid crest is absent in Balaenula astensis but this statement may be easily questioned. It would be better to state that in Balaenula astensis the supramastoid crest is reduced. The same character should be reviewed and changed accordingly in the character list in the Supplementary information file. Authors must check the English because their text is rich in syntax errors.

Description
Preservation: The authors use the term “corroded” which refers to chemical attack to the bone. Is that the correct term to be used here? If so, which kind of chemical attack occurred?
Physical maturity: check the English. Rephrase the very long last sentence to help understanding.
Supraoccipital: the references to the stages of suture fusion described by Welsh & Berta (2011) are for balaenopteroids. The authors must state that. As an example, on p. 23 line 388 they should write what follows: (corresponding to stage SR4 of balaenopteroids).
Basisphenoid-presphenoid: on p. 25 line 448, the authors used abbreviations to indicate length and width. I recommend writing length and width as abbreviations are not defined in the previous lines or in the Methods section.
Pterygoid: authors used ~ or ≈ inconsistently throughout the manuscript. I recommend being consistent.
Frontal: p. 31 line 588: how can the authors explain the presence of the rugose surface on the orbit (a character of calves) in an adult or subadult individual?
Periotic: p. 34 line 648: what about the tractus spiralis foraminosus?

Phylogenetic analysis
Check the English carefully. P. 35 line 663: please, define the posterior border of the zygomatic process (of the squamosal, I suppose). P. 37 line 708: note that in Figs 2 and 3 of Bisconti (2000), the narial process of the frontal of Balaenula astensis is absent thus it is impossible to accept it as a synapomorphy of the clade including Balaenula astensis, Balaenella, balaena and Eubalaena. Moreover, it the presence of such a process is questionable in Eubalaena australis based on several illustrations. The analysis results also in the paraphyly of Balaenopteridae as Eschrichtiidae is included within it.
I suggest moving the Comparative skull anatomy before the Phylogenetic analysis as it provides evidence used to code character states and, in the end, for supporting nodes in the cladograms.

Discussion
The authors make comparisons to an unpublished skull (MPL 5-21) and use characters of such a skull as evidence against the monophyly of Balaenoidea. However, they must illustrate what they are referring to otherwise their evidence cannot be accepted. I recommend illustrating the characters they are describing in the text or remove the text about MPL 5-21 to avoid ambiguity. I must note that the character under discussion in this paragraph is related to the development of the coronoid process of the dentary. It must be remarked that the use and codings of ontogenetic characters should be carefully discussed as in many cases may be problematic. Panchen (1994 in Hall, Homoplasy, Academic Press) strongly recommend to use adult morphologies to establish homology and cites authoritative references. I agree that ontogenetic data may be used but only in cases in which complete developmental paths are known but this is not the case of Balaenidae and Neobalaenidae.
Orientation of the skull: I recommend to remove this text unless the characters discussed in the Methods are better defined.
On p. 43 the authors use the italic for a title. Is this correct?
P. 43 line 866: anterior process of the periotic? Please, specify to which bone belong the characters mentioned in this paragraph. Moreover, this whole paragraph is to be carefully revised, as the English is inadequate and rich in misspellings and wrong syntax.
P. 46 line 918 and followings: Pyenson & Sponberg (2011) applied their methodology to the reconstruction of two extinct cetaceans including one balaenopterid (Balaenoptera siberi). In their paper, they are honest and clear in stating that the bizygomatic width overestimated the total length of B. siberi and that corrections are to be made in order to obtain the true length of the specimen under discussion. In this sense, the methods of Pyenson & Sponberg (2011) must be used carefully and with full complement of statistical corrections, they suggested. Finally, they developed specific regression equations to predict the total body length of specific clades of mysticetes: stem mysticetes and Balaenopteroidea. They did not develop specific equations for Balaenidae and, judging from their Fig. 3, stem mysticetes do not include Balaenidae. For these reasons, the predicted body sizes of the present manuscript cannot be accepted.

Figures
Figure 4. The supraoccipital fossa should be named external occipital fossa according to the text.
Figure 5. The lateral border of the exoccipital seems wrongly placed. In fact, in Fig. 5A (right part), such a border corresponds to the lateral edge of the sternomastoid fossa which is not the case, in my opinion. A thinner line located more dorsally seems to better correspond to the lateral border of the exoccipital. Moreover, it is possible to indicate the posterior process exposure in the right part of Fig. 5A. In Fig. 5B the authors should indicate the parietal-frontal suture and the squamosal-exoccipital suture which are clearly visible.
Figure 6B. The authors should indicate the parietal-frontal suture. The lateral border of the exoccipital indicated in this figure does not agree with that in Fig. 5. Please, decide and correct.
Figure 7. Postglenoid must be changed into postglenoid process.
Figure 11. Proximal opening of the facial canal must be changed into endocranial opening of the facial canal according to Mead & Fordyce (2009) which is the reference cited by the authors.
Figure 13. In the reconstruction of the skull of Balaenella brachyrhynus the shape of the premaxilla is different from the photographic representation provided in the original description. In the latter, there is a sharp edge between the anterior border of the narial fossa in lateral view and the portion of the rostrum that is anterior to such an edge has a different orientation. In the authors’ reconstruction, the horizontal portion of the premaxilla is too long if compared to the shape of the premaxilla of the original description and the change in orientation of the premaxilla seems not as marked as in the original description. I recommend the authors to provide a better reconstruction of Balaenella brachyrhynus showing the right details of the rostrum. From a different viewpoint, I would recommend the authors to remove this illustration unless they provide better-grounded characters for getting correct skull orientations for Balaenidae (see above).

·

Basic reporting

See below in general comments.

Experimental design

See below in general comments.

Validity of the findings

See below in general comments.

Additional comments

I am very pleased to see this manuscript. A more proper description and thorough analysis of Morenocetus parvus was long overdue. This manuscript does both things, which makes this a much needed contribution to our understanding of marine mammal diversity from South America, and the evolutionary history of balaenids. This work will no doubt be of interest to paleontologist, neontologist as well as anyone interested in the evolutionary history of baleen whales.

The manuscript is generally well written and I commend the authors on the high quality images, illustrations and analysis. However, I do have some minor concerns that are outlined in detail below and that I hope the authors take into consideration.

Specific comments:

Lines 87-89: I suggest changing the beginning of this sentence from “In latest Bisconti’s (2005) analysis…” to “In the latest analysis (Bisconti, 2005)…”

Lines 182-184: I suggest the authors to be cautious when using the implied weight method as it may not necessarily result in the best topology. See the papers by Congreve & Lamsdell (2016) and Puttick et al. (2017) where they discuss the inconsistencies and lack of accuracy of using implied weights. For example see the different position of Mauicetus parki in Fig. 12B, as the sister taxon of Plicogulae and not as a balaenopteroid as in previous analyses (e.g. Marx & Fordyce, 2015). See also Tsai & Fordyce (2016) which show similar discrepancies with this taxon in the equal vs implied weight results.

Congreve, C.R. &. J.C. Lamsdell. 2016. Implied weighthing and its utility in paleontological datasets: a study using modeled phylogenetic matrices. Paleontology 59:447-462.

Puttick, M.N., J.E. O’Reilly, A.R. Tanner, J.F. Fleming, J. Clark, L. Holloway, J. Lozano-Fernandez, L.A. Parry, J.E. Tarver, D. Pisani & P.C.J. Donoghue. 2017. Uncertain-tree: discriminating among competing approaches to the phylogenetic analysis of phenotype data. Proceedings of the Royal Society B 284:20162290.

Line 275: Is there a full name for Cremonesi?

Line 290: I suggest changing “… and lateral edges of supraoccipital straight.” to “… straight lateral edges of the supraoccipital.”

Line 312: correct typo here “… having ths supraorbital…”

Line 315: “thicked orbital rim…” did the authors meant “thicker orbital rim…”

Lines 379-382: In figure 4B the “external occipital fossa” is labeled as “Supraoccipital fossa” please update the figure (or text) so that they are consistent.

Lines 567-569: There seems to be a discrepancy with the first half of this sentence “The supraorbital process is short transversely and long anteroposteriorly:” however, it looks more like the supraorbital process is long transversely, and short anteroposteriorly, not the other way around.

Line 770: period missing after Eubalaena spp.

Line 782: add period after Balaenula sp. and only italize genus

Lines 792-793: separate “Discussion…” section from preceding paragraph.

Line 876: remove space between Chaeomysticeti and comma

Line 887: remove space between “stasis/” and “morphological”

Line 893: I suggest changing “Mysticetes” to “mysticetes”

Line 912: montalionis is not completely italized

Line 914: typo at end of line “extinc”

Line 915: typo, change “diferent” to “different”

Lines 927-930: see also Slater et al. 2017

·

Basic reporting

Some minor grammatical and spelling issues are present and the text would benefit from some language editing in places.

Experimental design

Se below.

Validity of the findings

See below.

Additional comments

This manuscript by Buono et al. provides a long-awaited redescription of the early right whale Morenocetus parvus from the lower Miocene of Argentina. Right whales have often been interpreted as being the most conservative extant baleen whales owing to this species and its early occurrence, predating most other families by nearly 10 My. This paper includes a well-written, detailed, and long overdue morphological description of Morenocetus as well as a reconsideration of the history of certain hallmark anatomical features of balaenids and a phylogenetic analysis.

I have no major concerns with the manuscript and think that it deserves publication. I have a list of minor comments below, and suggest minor revision. Some English language editing is necessary for some sections - but note that with a few exceptions, the language does not seem to impact clarity.

Congratulations to Monica and colleagues on getting this manuscript completed!

Kind regards, R.W. Boessenecker, Ph.D.

46: change to "Balaenids have been considered"
51: replace cf with "in comparison to"
53: Extremely large head: perhaps give some brief statements on cranial length as a percentage of body length
54: length of baleen plates in meters?
69: confirmed by whom? In the spirit of thoroughness, a few citations for previous phylogenetic analyses to have included Morenocetus would help.
120: is MLP 5-21 Pleistocene or Holocene instead? Be specific.
135-136: "to avoid its destruction" do the authors mean that removing the periotic would cause irreparable damage, either to the earbone, or the skull?
141: delete the comma after ’specimens'
142: change "based" to "description focused"
171: "list of characters was modified..." how? how many characters were removed? how many added? why? A separate section of text outlining and justifying these changes is necessary.
199: could BZW be estimated and scaled based on ratios of BZW to exoccipital width in other balaenids?
Geologic setting: I thank the authors for including an informative stratigraphic column and not shying away from geological context.
280: narrow exposure; report as a % of exoccipital width?
281: change deeply to deep
288: quantify this as a % of postorbital or exoccipital width.
292: change to "crest-like parieto-squamosal suture"
297: narial process of frontal
299: lateral tuberosity of the periotic? please clarify
304: lateral hypertrophy of the body of the periotic?
307: supramastoid crest extending all the way to tip of zygomatic? please be more specific about the condition in Eubalaena and Balaena.
312: Differs; "ths" to "the"
313: frontal gradually sloping away ventrolaterally
314: half of distance: unclear, please clarify.
373: does not reach the level of the preorbital process?
386: approximately
388: delete the extra comma after 774.
442: Which neck flexor muscles? Be specific - Schulte 1916 might be a good enough reference.
466: hamuli
473: in comparison to what other taxon?
481: I think there's an extra space between 774 and the comma
498: alisphenoid
517: I notice that the authors use both "periotic fossa" and the problematic term cranial hiatus. According to Mead and Fordyce 2009 the latter term is poorly defined and they introduced the term periotic fossa to replace it; are the authors using it in the correct sense, or as the pit that the periotic sits within?
527: Should an unpublished thesis be cited here? I personally don't think so.
562: I strongly think that the "frontal ridge" proposed here is the orbitotemporal crest, which has shifted anteriorly within Crown Mysticeti. Many stem balaenopteroids like Parietobalaena and Pelocetus have an orbitotemporal crest shifted anteriorly on the frontal. It's been a while since I have read relevant literature on muscle insertions on mysticete crania, but I recall that a dissection of Caperea prior to the 2013 Marine Mammal conference in Dunedin revealed that the temporalis inserted upon the entire dorsal surface of the supraorbital process of the frontal, terminating at a low ridge at the same position as what I have identified as the orbitotemporal crest.
577: change to "in extant balaenids"
586: dorsally concave?
615: change in to "to be"
626: obtuse angle - please quantify
684-685: suprameatal part of periotic not hypertrophied - Morenocetus appears to still have quite a large body of the periotic. This should be quantified, perhaps as a maximum transverse width expressed as a percentage of pars cochlearis width. Relative to an eomysticetid, the body of the periotic is enormous.
687: change specialized to derived?
718: unclear
812: A relevant citation that should probably be cited here for correct skull orientation as far as interpreting skull morphology is Yamada et al. 2006: Memoirs of the National Science Museum, Tokyo 44:1-10.
865: change to "exhibits"
876: derivate to derived
878: change to "hypertrophied body of periotic"
890: change to "major mysticete clades"
893-894: "this..." please rephrase.
894: Aquitanian or Burdigalian?
897-898: Unclear. Figure 13 is a picture of skulls; does this refer to Fig. 14? What is the taxonomic inclusion of Balaenopteroidea? Which balaenopteroids are present in the Chattian and Aquitanian? Some citations of specific examples should be provided in the text since these questions arise from lack of clarity in the figure and caption.
900: occur
942: The authors should include a table of Aquitanian and Burdigalian mysticetes, summarizing the lower Miocene record.
950: citations needed.
952: Paleobiology database is not a reasonable source to cite. Please cite published articles.
946-952: A recent paper by RE Fordyce and myself reports an eomysticetid from the lowermost Miocene of NZ and makes some of the same statements provided here in the "Pending Issues" section, and may be a worthwhile citation. RW Boessenecker and RE Fordyce, 2017. Cosmopolitanism and Miocene survival of Eomysticetidae (Cetacea: Mysticeti) revealed by new fossils from New Zealand. New Zealand Journal of Geology and Geophysics DOI: 10.1080/00288306.2017.1300176

---

## Round 0.2 · Minor Revisions

This manuscript was reviewed again, and two reviewers recommended acceptance, a third major revision. That reviewer, Reviewer 1, clearly read the manuscript very carefully and their suggestions fell into different categories.

Some of these flow from their own, unpublished, study of fossil mysticetes. I do not believe that changes based on this are needed, although it is important to be explicit about observations made in this manuscript. This concerns point 4 in their review.

Other suggestions from Reviewer 1 concern discussions, citations and references to published literature and records of observations on specimens listed . These suggestions should be considered and modifications made. This concerns points 1, 3, 5, 6, 7, 10, 15, 16, and 17 in their review.

A third kind of suggestions that were made by Reviewer 1 can be ignored in my view. This concerns points 2, 8, 9, 12, 13, 14, and 19 in their review.

Minor grammatical, spelling changes, such as point 11, 18

Reviewer 1 ·

Basic reporting

Professional English used throughout. Methods and discussion not completely unambiguous. Important changes to be made as in the General comments for the authors.
Literature references are mainly correct. Some additional citations needed as suggested . Field background and context are provided correctly in most of the cases but with some necessary changes to be made.
The article structure is OK. Some problems with a few figs as suggested in the General comments for the authors.
The paper is self-contained. Relationships between hypotheses and results to be revised by the authors.

Experimental design

Original primary research within aims and scope of the journal.
Research question well defined, relevant and meaningful. The paper fills an important knowledge gap as a redescription of Morenocetus is very welcome task in the field.
The investigation is mostly rigorous but fails in a number of points as discussed in the General comments to the authors.
Methods are described with sufficient detail about some aspects. Other aspects need clarifications.

Validity of the findings

The description of Morenocetus is surely an important and valid finding. Anatomical data are robust and controlled. Other aspects need clarification. See General comments to the authors.
Conclusion are well stated as far as anatomy and phylogeny are concerned. Other aspects (i.e., reconstruction of body size) to be clarified and discussed in better way.

Additional comments

I think that the a new description of Morenocetus parvus would be very welcome. All the students of mysticete phylogenetics would be happy to have such a description made in modern terms and with current criteria. Unfortunately, there is a number of issues that have to be made clear or that need additional scientific research. I will be more explicit below. Moreover, the newly added texts required a new review round that resulted in the need for further clarification in many points. I am sorry if this will delay the publication of this manuscript but I honestly feel the changes I request are really necessary. I will try to be highly specific in my review and will provide also illustrations to better express my thoughts.
Review of “The early Miocene balaenid Morenocetus parvus from Patagonia (Argentina) and the evolution of right whales” by Mónica R. Buono, Marta S. Fernández, Mario A. Cozzuol, Jose I. Cuitiño, and Erich M. G. Fitzgerald

The manuscript is now improved as far as the English is concerned and the new details added provide better quality and clarity. There are still some parts of the manuscript that, in my opinion, need to be changed in order to better present the scientific content and to ameliorate readers’ understanding. I will discuss these below. The line numbers are based on the Word file with tracked changes.
1. A new paper was recently published dealing with a revision of “Balaena” belgica that established the new species Eubalaena ianitrix (Bisconti et al. 2017, PeerJ). The authors cited such a paper several times in the text but they still use “Balaena” belgica in their text. Unless they can provide a criticism to the taxonomy of Bisconti et al. (2017), they must change “Balaena” belgica into Eubalaena ianitrix throughout the text and in Fig. 12.
2. In the differential diagnosis, the authors must make comparisons to Eubalaena ianitrix.
3. In Fig. 12, the authors must change Eschrichtius gastaldii into Eschrichtioides gastaldii unless they can justify their assignment to Eschrichtius.
4. In Fig. 13, the premaxilla of Balaenella is still drawn imprecisely. I attach an image for giving the authors an idea of what I am saying (Fig. R1). Moreover, it is not clear how they reconstructed the length of the rostrum of this species given that its anteriormost portion is not preserved. Finally, the dorsoventral diameter of the maxilla in lateral view approaching the lateral process is undersized with respect to the photographs published in the original description. Based on that, the reconstruction of this skull is not done thoroughly, in my opinion, and should be done more precisely or removed.
5. Line 90. It is not clear to me why the authors did not cite also Bisconti (2000, Palaeontographia Italica, 87: 37-66), Churchill et al. (2012, Marine Mammal Science, 28: 497-521), and Bisconti et al. (2017, PeerJ) in the comprehensive analyses of mysticete phylogeny focusing on balaenids.
6. Lines 171-175. To be honest, I did not find any reference to the specimens you dissected or analysed to understand the pattern of sutural closure in balaenids in the Supplementary Information. You should provide (at least in the form of a table in the Supplementary Information) the specimens you analysed (species, repository and number), their ages and the pattern of sutural closure observed in each of them. I note that Welsh and Berta dedicated a whole paper to this subject while you are discussing it very quickly and without providing the necessary information. Otherwise, we simply have to trust you. We could do that but it is not a scientific procedure in a strict sense.
7. Line 206. In the revised text, the authors write about “dissection of extant balaenids”. What balaenids? From the manuscript, it seems that only specimens of Eubalaena australis were dissected. Is this correct? What about the other species of Eubalaena? What about Balaena mysticetus? It seems that only character 1 was tested with Balaena mysticetus. If the authors do not have data about these species, they should carefully apply their method to the fossil specimens as a significant taxonomic sample of living balaenids was not tested. I strongly recommend to use a more dubitative textual communication approach undermining that your results (regarding the orientation of the skull) are acceptable if the landmarks can be applied to all the living balaenids at least. Unfortunately, currently that is not known and the resulting orientations may be not as precise as desirable. I guess that this point will be the focus of criticisms by other authors if published as is, especially because the authors are very bold about their own results.
8. Line 211. accurate reconstruction of the branching order of the sister groups may influence the reconstruction of character state evolution. The authors have to be aware that their work results in a hypothesis of phylogenetic relationships of Mysticeti with focus on Balaenidae. They decided to order some characters thus suggesting that a number of evolutionary trajectories are known (which should be demonstrated). Thus they accept that Balaenopteridae and Eschrichtiidae are merged together. They have to remark this in the text.
9. Line 226. It is a pity that the present authors pruned the original dataset as Wagner (2000, Evolution, 54: 365-386) and Heath et al. (2008, Journal of Systematics and Evolution, 46: 239-257) convincingly argued that more taxa are highly useful in the reconstruction of phylogenies in groups whose diversity is mainly formed by extinct taxa. Including more taxa should increase the probability to retrieve the correct phylogeny. I strongly suggest to include the whole dataset by McGowen et al. (2008) and Marx and Fordyce (2015).
10. Line 352. Actually, there are three different specimens of Balaenula that have been assigned to two different species and to an unnamed taxon. These are Balaenula balaenopsis, Balaenula astensis and the Balaenula sp. from Japan. As the taxonomy of these three specimens is debated, in the diagnosis, the authors should state clearly which is the Balaenula they are comparing to Morenocetus or they should state which definion of Balaenula they are referring to.
11. Line 665. Change disecction in dissection.
12. In the comparative skull morphology, comparisons to Caperea and Miocaperea would be useful because Morenocetus resembles these species closely in the high position of the orbit, in the lateral exposure of the posterior process of the periotic and in some aspects of the supraoccipital morphology (especially the anterior border).
13. Line 888. The authors shows that the narial process of the frontal is present also in Balaenula astensis and use this character as a synapomorphy of a larger clade. I attach here a figure illustrating the different morphologies of the narial process of the frontal in Balaenula astensis, Balaenella brachyrhynus and Balaena mysticetus supporting the view that these three taxa exhibit at least two different morphologies of this character (Fig. R2). Based on that, I suggest that the authors have to subdivide the character about the narial process into two different characters: 1) presence/absence of the narial process; 2) relative development of the narial process (short/elongate). In this way, they can correctly represent the morphological diversity of this feature in balaenids.
14. Line 902. To my knowledge, based on direct examination of several specimens, the crest at the parietal-squamosal suture is not present in Eubalaena glacialis and Balaenula astensis. Is this proposed synapomorphy a result of character optimization made by the computer program or the authors actually found such a crest in specimens of these species? Being positive the second option, please, cite repository and numbers of the specimens because that would be very useful to other authors.
15. Line 1127. Actually, Balaenella is not a historical form as it was described for the first time in 2005. Balaenula astensis received two different revisions in 2000 and 2003 based on computer-assisted phylogenetic analyses and morphological re-descriptions in modern terms. It is evident that the authors do not agree with these descriptions and revisions, and this is OK, of course. However, in my opinion, they should re-phrase this last sentence in order to avoid to be in error by considering Balaenella a historical form. I suggest them to mention Balaenula balaenopsis and Balaena primigenia, as examples of historical forms.
16. In the Supplementary Information, List of the specimens studied for the anatomical and phylogenetic analysis, please, note that Balaenula astensis preserves both periotics and both tympanic bullae (Bisconti 2003, Cranium 20: 9-50).
17. The same list is presented in an inconsistent way: some specimens are associated to a list of materials while others are merely cited. A consistently compiled list would be more useful to understand what exactly you examined.
18. Literature cited in Supplementary Information not in alphabetical order. The authors have to write it in alphabetical order.
19. The last point I want to discuss is about the reconstruction of the body size of Morenocetus and, more generally, the way the authors discuss their work and the work of other authors. In fact, in my opinion, this is a case in which the discussion is made in a misleading way and gives the readers ambiguous information.
The authors insist that the method developed by Pyenson and Sponberg (2011) is the best available method at the moment. This is certainly true about the reconstruction of body size in basal mysticetes and balaenopteroids; this is certainly not true about balaenids. In fact, such a method consists in an regression equation developed from a large dataset of mysticete measurements that, unfortunately, does not include a single balaenid specimen. As a wealth of published literature recognizes that the body proportions of balaenids are different from the other mysticete families, it is obvious that one should not expect that such an equation can retrieve accurate results for balaenids. In the introduction of the present manuscript (line 54 in the newly added text), the authors acknowledge that one of the characterizing features of living balaenids is their body proportions in which the head is about one-third of the total body length. This character is not observed in any other mysticete. The head length:body length ratio is based on a large number of measurements available in early literature and summarized in more recent papers (Cummings 1985, Handbooks of marine mammals, vol. 3, pp. 275-304, Academic Press; Reeves and Leatherwood 1985, Handbooks of marine mammals, vol. 3, pp. 305-344, Academic Press) and is usually accepted as a matter of fact.
I appreciate that the authors of the present manuscript recognized that the Pyenson and Sponberg (2011) equation was not specifically developed for balaenids (line 247). However, I strongly disagree with the statement in the rebuttal letter in which the authors state that “there is no basis for assuming that ‘the results are expected to be wrong’”. In fact, the results are expected to be wrong because: (1) living balaenids (and we can measure only complete skeletons from the living species) have different head:body length ratio from other mysticetes and (2) the Pyenson and Sponberg equation is not based on balaenids. Thus, the Pyenson and Sponberg equation provides results compatible with an expected head:body length ratio different from that of balaenids.
In the rebuttal letter, the authors write also that “this equation was recently applied with success to Balaenidae in a major synthesis of mysticete body size evolution (Slater, Goldbogen and Pyenson 2017)”. However, Slater et al. (2017) did not provide the original dataset of the measurements they used to base the equation used in their paper; they provided only a table with the estimated body lengths of a number of mysticetes. Therefore, it is not possible to know if their equation is based on a dataset including balaenids or not. In this sense, their reconstructions of the body sizes of balaenids is hard to accept as presented in such a paper. In the end, the use of that equation in other papers is not per se a source of support for the validity of the equation itself.
In my opinion, the authors may publish their results in the way they want bearing in mind that this means that their paper will be prone to strong criticism in future publications by other authors.
Another aspect of the text about the reconstruction of Morenocetus body size is around the discussion of the paper by Bisconti et al. (2017). From the present manuscript, it seems that Bisconti et al. (2017) used alternative methods to the Pyenson and Sponberg equation and that they accepted the results of these methods without discussion. This does not correspond to truth. If the authors read carefully the paper, they can find that Bisconti et al. (2017) used several methods for inferring the body length of Eubalaena ianitrix including the equation of Pyenson and Sponberg (2011). Moreover, Bisconti et al. (2017) discussed pros and cons of each method and concluded that none of the available methods is exempt from problems. Bisconti et al. (2017) acknowledged that their reconstruction of the body mass was probably wrong and discussed their results in a broad sense. The harsh criticism to the paper of Bisconti et al. (2017) made by the present authors does not add anything to the discussion of the methodologies about body size reconstruction in fossil balaenids and misleads the readers by ignoring the real content of the work published by Bisconti et al. (2017). I strongly recommend to rephrase the discussion of the Bisconti et al. (2017) work from line 1067 onwards in order to correctly represent the real content of that paper.
It is also possible that the application of the Pyenson and Sponberg (2011) equation by Bisconti et al. (2017) was done better than in the present manuscript as Bisconti et al. (2017) made the corrections suggested by Pyenson and Sponberg (2011), i.e. they reduced the results by 47% and 37% in order to get a range of body size variation, while the present authors present only one result. Is the value obtained by the present authors corrected as suggested by Pyenson and Sponberg or not? To be honest, I was unable to replicate the results of the present authors when I checked the results presented in the present manuscript. In fact, the application of the equation of Pyenson and Sponberg to a bizygomatic width of 570 mm allowed me to find a total body length of 7.46 m. After the changes suggested by Pyenson and Sponberg (i.e., reduction of 47% and 37% of the initial result), I found two body lengths for Morenocetus: 3.96 m and 4.7 m that are different from the results of the present manuscript (5.2 m). The application of the same equation to a bizygomatic width of 620 mm (specimen MLP 5-15) resulted in a total body length of 8.06 m and, after the corrections, in a body length range included between 4.26 and 5.16 m that is different from the result of the present manuscript (5.6 m). In light of these new results, the discrepancies with the Bisconti et al. reconstruction of the body length of Eubalaena ianitrix seems more reasonable: 3.96-4.7 m for Morenocetus and 5-7 m for E. ianitrix which is almost twice of Morenocetus.
By the way, I noticed that the reconstructions of the body lengths of other fossil balaenids made by the present authors by using the equation of Pyenson and Sponberg differ from the published reconstructions made by Slater et al. (2017, see their supplementary information). In fact, in the present manuscript, the total body length of Balaenella brachyrhynus is 7.9 m while Slater et al. found that this whale was 7.2 m in length; Balaena montalionis is 7.2 m in this manuscript but the reconstruction made by Slater et al. found this whale being 4.87 m. Moreover, judging from published measurements, Balaenella brachyrhynus seems smaller than Balaenula astensis. In fact, the supraoccipital length of B. brachyrhynus is 310 mm (Bisconti 2005) and that of Balaenula astensis is 350 mm; the maximum diameter of the foramen magnum of Balaenella brachyrhynus is 64 mm and that of Balaenula astensis is 75 mm. How can Balaenella brachyrhynus be longer than Balaenula astensis in total body length? In fact, in the reconstruction of the present manuscript, the total body length of Balaenula astensis is 6.6 m and that of Balaenella brachyrhynus is 7.9 m.
I cannot understand these discrepancies and strongly recommend the present authors to calculate again the body length of Morenocetus and other balaenid species and, eventually, to discuss these discrepancies in the Discussion section.

In my opinion, due to the above observations, the manuscript is still in need of a serious revision. Figures attached to this review are provided in a separate file.

Annotated reviews are not available for download in order to protect the identity of reviewers who chose to remain anonymous.

·

Basic reporting

See below in general comments.

Experimental design

See below in general comments.

Validity of the findings

See below in general comments.

Additional comments

I am very pleased to see the revised version of this manuscript. The authors have properly addressed all the issues brought up by myself and the other reviewers. As stated in the first review of this work, a more proper description and thorough analysis of Morenocetus parvus was long overdue. This manuscript does both things, which makes this a much-needed contribution to our understanding of marine mammal diversity from South America, and the evolutionary history of Balaenidae.

The whole discussion of the orientation of the skull of balaenids is very helpful and will hopefully be followed by anyone else working on this group. The authors did an outstanding job in the description of the specimens and related figures. Furthermore, the detailed discussion on the morphology and relationships of Morenocetus parvus and Balaenidae is helpful and thorough, especially making the tie-in between the text and the character/state used in the phylogenetic analysis. The manuscript is well written and I commend the authors on the high quality images and illustrations (which are extremely helpful and detailed). This work is no doubt of interest to paleontologist, neontologist as well as anyone interested in the evolutionary history of baleen whales.

·

Basic reporting

OK

Experimental design

OK

Validity of the findings

OK

Additional comments

The authors have done an excellent job in addressing all of my comments and I recommend acceptance of the paper - and look forward to seeing it in print. Kind regards, R.W. Boessenecker

---

## Round 0.3 · accepted · Accept

Thank you for your diligence is evaluating the comments by the reviewers. I believe that you have seriously considered all of them, and have clarified some issues that have improved the manuscript.